# Interactions between metabolism and growth can determine the co-existence of *Staphylococcus aureus* and *Pseudomonas aeruginosa*

Camryn Pajon[1†], Marla C Fortoul[1†], Gabriela Diaz-Tang[1], Estefania Marin Meneses[1], Ariane R Kalifa[1], Elinor Sevy[1], Taniya Mariah[1], Brandon Toscan[1], Maili Marcelin[1], Daniella M Hernandez[1], Melissa M Marzouk[1], Allison J Lopatkin[2,3,4], Omar Tonsi Eldakar[1], Robert P Smith[1,5]*

[1]Department of Biological Sciences, Halmos College of Arts and Sciences, Nova Southeastern University, Fort Lauderdale, United States; [2]Department of Biology, Barnard College, Columbia University, New York, United States; [3]Data Science Institute, Columbia University, New York, United States; [4]Department of Ecology, Evolution, and Environmental Biology, Columbia University, New York, United States; [5]Cell Therapy Institute, Kiran Patel College of Allopathic Medicine, Nova Southeastern University, Fort Lauderdale, United States

**\*For correspondence:**
rsmith@nova.edu

[†]These authors contributed equally to this work

**Competing interest:** The authors declare that no competing interests exist.

**Abstract** Most bacteria exist and interact within polymicrobial communities. These interactions produce unique compounds, increase virulence and augment antibiotic resistance. One community associated with negative healthcare outcomes consists of *Pseudomonas aeruginosa* and *Staphylococcus aureus*. When co-cultured, virulence factors secreted by *P. aeruginosa* reduce metabolism and growth in *S. aureus*. When grown in vitro, this allows *P. aeruginosa* to drive *S. aureus* toward extinction. However, when found in vivo, both species can co-exist. Previous work has noted that this may be due to altered gene expression or mutations. However, little is known about how the growth environment could influence the co-existence of both species. Using a combination of mathematical modeling and experimentation, we show that changes to bacterial growth and metabolism caused by differences in the growth environment can determine the final population composition. We found that changing the carbon source in growth media affects the ratio of ATP to growth rate for both species, a metric we call absolute growth. We found that as a growth environment increases the absolute growth for one species, that species will increasingly dominate the co-culture. This is due to interactions between growth, metabolism, and metabolism-altering virulence factors produced by *P. aeruginosa*. Finally, we show that the relationship between absolute growth and the final population composition can be perturbed by altering the spatial structure in the community. Our results demonstrate that differences in growth environment can account for conflicting observations regarding the co-existence of these bacterial species in the literature, provides support for the intermediate disturbance hypothesis, and may offer a novel mechanism to manipulate polymicrobial populations.

## Editor's evaluation

How the pathogens *Pseudomonas* aeruginosa and *Staphylococcus aureus* compete and co-occur within opportunistic infections is an important problem, but the major drivers of these interactions remain unclear. Here the authors contribute a fundamental advance by defining parameters that predict the coexistence of these microbes using their absolute growth in various nutritional

conditions, which could explain the dominance of one or the other during infections. Within a defined context, this valuable study provides solid support for a novel framework in which to evaluate this clinically important species interaction.

## Introduction

Microbes rarely exist in isolation. It is more common to find them as part of diverse polymicrobial communities comprised of multiple microbial species (e.g. *Peters et al., 2012*). Interactions within these communities can be synergistic or antagonistic. While cooperation can augment the stability of the community (*Cavaliere et al., 2017*), competition amongst community members can stimulate the expression of gene products in an effort to drive competitors toward extinction (*Burgess et al., 1999*). In both cases, these interactions can produce novel behaviors that are not otherwise observed when community members are grown in isolation. On the one hand, they can have beneficial functions; they can produce useful products, such as biofuels (*Mittermeier et al., 2023*), and degrade harmful environmental pollutants (*Brune and Bayer, 2012*). On the other hand, they can produce behaviors detrimental to human welfare, including augmented virulence (*Stacy et al., 2014*) and increased antibiotic resistance (*Perez et al., 2014*). Accordingly, understanding the general principles that shape the interactions and co-existence of members within a polymicrobial community has wide-ranging implications, such as enhancing the production of beneficial products, and strategies to attenuate antibiotic resistance.

One polymicrobial community that is commonly observed in the clinic and is of increasing concern owing to the emergence of antibiotic resistant strains is composed of *Pseudomonas aeruginosa* and *Staphylococcus aureus*. These bacteria co-colonize burns (*Norbury et al., 2016*), individuals with cystic fibrosis (*Sagel et al., 2009*), and chronic infection sites (*Serra et al., 2015*; *Pastar et al., 2013*). Their co-occurrence leads to significantly worse healthcare outcomes (*Hotterbeekx et al., 2017*; *Maliniak et al., 2016*), including increased inflammation during pneumonia (*Sagel et al., 2009*) and increased wound healing time (*Pastar et al., 2013*; *Dalton et al., 2011*; *Seth et al., 2012*). The interactions between *S. aureus* and *P. aeruginosa* are well understood and, at a fundamental level, influence the growth and metabolism of both species (*Figure 1A*). Their co-culture induces interspecies competition, which enhances antibiotic resistance and virulence. In the presence of *P. aeruginosa*, *S. aureus* upregulates the expression of virulence factors, including staphylococcal protein A, α-hemolysis and Panton-Valentine leucocidin (*Pastar et al., 2013*). While *S. aureus* can initially aid in the establishment of the *P. aeruginosa* population (*Alves et al., 2018*), production of N-acetylglucosamine from *S. aureus* augments the virulence of *P. aeruginosa* by upregulating expression of multiple secreted virulence factors. Many of these are regulated by quorum sensing and include pyocyanin (*Korgaonkar and Whiteley, 2011*), pyoverdine (*Orazi et al., 2019*), 2-Heptyl-4-hydroxyquinoline N-oxide (*Machan et al., 1992*), and 4-hydroxy-2-heptylquinoline N-oxide (HQNO) (*Li et al., 2015*; *Déziel et al., 2004*; *Nadal Jimenez et al., 2012*; *Korgaonkar et al., 2013*). Some of these virulence factors reduce metabolism in *S. aureus* through multiple mechanisms. Pyocyanin induces the production of reactive oxygen species in *S. aureus* by reacting with menadione, which results in oxidative damage (*Noto et al., 2017*). Pyocyanin has been found to interfere with electron transport, reducing respiration (*Biswas et al., 2009*). The siderophores pyoverdine and pyochelin drive *S. aureus* into fermentation (*Filkins et al., 2015*). While exposure to HQNO can initially promote biofilm formation in *S. aureus* (*Fugère et al., 2014*), it eventually interferes with the activity of cytochrome b, which reduces ATP production (*Nguyen and Oglesby-Sherrouse, 2016*). N-3-oxo-dodecanoyl-L-homoserine lactone (3OC12HSL) produced by *P. aeruginosa* for the purposes of quorum sensing has been shown to reduce the growth of *S. aureus* (*Qazi et al., 2006*). Overall, multiple virulence factors interfere with growth and metabolism in *S. aureus*, ultimately leading to facultative respiration and increased antibiotic tolerance (*DeLeon et al., 2014*). Without a fast-growing competitor, and with additional nutrients released by *S. aureus* (*Mashburn et al., 2005*), *P. aeruginosa* can increase its growth and density (*Figure 1A*). Overall, co-occurrence of *P. aeruginosa* and *S. aureus* alters growth and metabolism, which ultimately results in a highly virulent and antibiotic-resistant infection.

Previous studies have reported the co-existence dynamics of *S. aureus* and *P. aeruginosa* in human hosts. *S. aureus* is the predominant bacterial species isolated from pediatric patients with cystic fibrosis. However, during adolescence and into adulthood, the frequency of *S. aureus* decreases, while

**eLife digest** Infections caused by multiple types of bacteria are tough to treat. For example, co-infections with *Staphylococcus aureus* and *Pseudomonas aeruginosa* are so difficult to cure they may persist for years in humans and cause serious illness. But when these two types of bacteria are grown together in the laboratory, *P. aeruginosa* kills off all the *S. aureus*.

Learning why these two types of bacteria can coexist in people but not in the laboratory may lead to new treatments to clear infections. It may also help scientists grow beneficial bacteria mixes that break down pollution or produce biofuels.

Pajon and Fortoul et al. show that interactions between bacterial metabolism and growth rate determine whether *S. aureus* and *P. aeruginosa* coexist. In the experiments, they grew both types of bacteria in different environments with different food sources. They measured their growth and metabolism and how many bacteria of each species survived over time. Then, they used their data to develop a mathematical model and tested its predictions in the laboratory again. The type of bacteria that had more energy also grew faster and outcompeted the other species. Measuring the growth rate of the two species allowed the scientists to predict which one would win out and what the tipping point would be. Physically disrupting the mix of bacteria disrupted this relationship.

These results may help explain what allows these bacteria to coexist in some settings but not others. It may enable scientists to develop new ways to treat infections with *P. aeruginosa* and *S. aureus* that work by manipulating growth in the two species.

Bacterial growth and metabolism are known to drive antibacterial resistance. Studies in mice using drugs or other therapies to manipulate growth and metabolism may help scientists thwart these resistance mechanisms. The results may also help scientists design and grow beneficial multispecies bacteria communities.

the frequency of *P. aeruginosa* increases (*Filkins et al., 2015*; *Rajan and Saiman, 2002*). One leading hypothesis to explain this trend is that *P. aeruginosa* outcompetes *S. aureus* through the production of HQNO and siderophores. Indeed, when co-cultured in vitro, *P. aeruginosa* has been shown to competitively exclude *S. aureus* over 24 hr (*Filkins et al., 2015*). However, long-term co-existence of these species has been observed in infected hosts, which has prompted investigations into mechanisms that can account for these conflicting observations. For example, overproduction of alginate by *P. aeruginosa* (*Limoli et al., 2017*; *Price et al., 2020*), upregulation of super-oxide dismutase (*Treffon et al., 2018*), decreased production of HQNO (*Frydenlund Michelsen et al., 2016*), and spatial segregation (*DeLeon et al., 2014*) are expected to facilitate co-existence. Differences in growth environments have also been suggested to play a role in facilitating co-existence. These include differences in environmental albumin concentration (*Smith et al., 2017*) and the production of proteins by the immune system (*Wakeman et al., 2016*). Interestingly, previous work has demonstrated that differences in nutrient availability in the growth environment can affect growth and metabolism of bacteria (*Pereira and Berry, 2017*). Indeed, the presence of different nutrients and metabolites in host microenvironment has been previously shown to impact metabolism and growth of bacteria (*Maslowski, 2019*; *Bjarnsholt et al., 2022*). Thus, it is possible that differences in nutrients that define the growth environment can affect the co-existence of *P. aeruginosa* and *S. aureus*. A growth environment that provides for a high metabolic rate in *S. aureus* could potentially buffer it against virulence factors produced by *P. aeruginosa* that reduce metabolism and growth. In turn, this could facilitate co-existence and perhaps allow *S. aureus* to outcompete *P. aeruginosa*. Alternatively, growth environments that reduce growth and metabolism of *S. aureus* could make that population more sensitive to these virulence factors, thus allowing dominance of *P. aeruginosa* and reducing co-existence. While these hypotheses are plausible, they have yet to be investigated. Thus, we ask whether growth environment-driven changes in growth rate and metabolism affect the co-existence of *P. aeruginosa* and *S. aureus* in co-culture. Addressing the role of growth environment on species interactions has implications in the bottom-up design of polymicrobial communities and may lead to novel treatment strategies in the clinic.

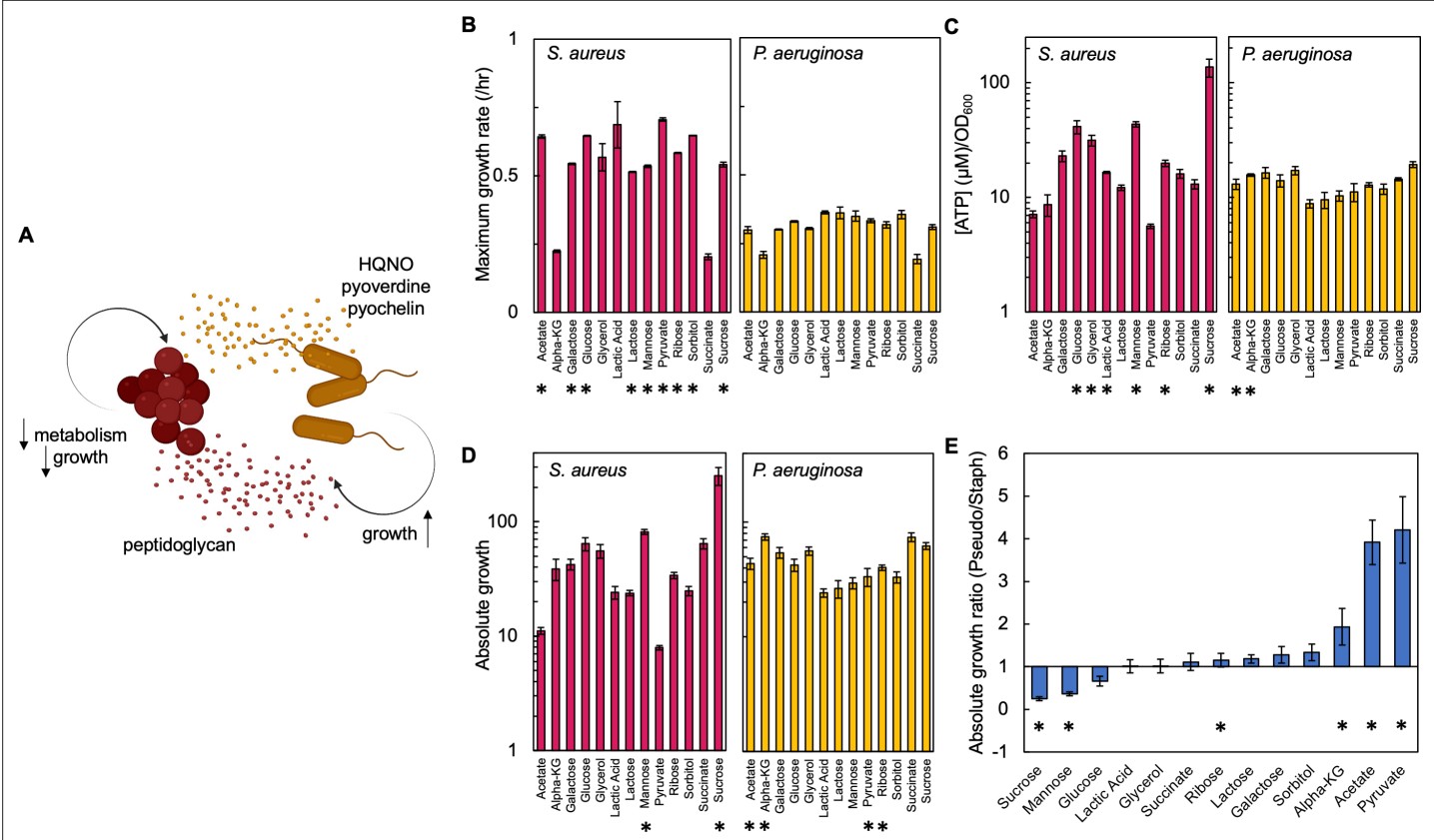

**Figure 1.** Carbon source in the growth medium affects growth and metabolism in *P. aeruginosa* and *S. aureus*; together this determines absolute growth. (**A**) Core interactions that affect growth and metabolism of *S. aureus* (red) and *P. aeruginosa* (yellow) when in co-culture. Peptidoglycan from *S. aureus* activates the expression of virulence factors from *P. aeruginosa*. These virulence factors, pyoverdine, pyochelin and HQNO, reduce metabolism and growth of *S. aureus*, which provides a benefit to *P. aeruginosa*. (**B**) Maximum growth rate of *S. aureus* (left) and *P. aeruginosa* (right) when grown in TSB medium with different carbon sources. Growth curves in *Figure 1—figure supplement 1*. Average from a minimum of three biological replicates. Error bars are standard error of the mean (SEM). * indicates a significantly greater maximum growth rate (two-tailed t-test, $P \leq 0.027$). For panels B, C, and D the exact number of biological replicates and all p values are in *Supplementary file 1*. (**C**) The concentration of ATP (µM) produced by *S. aureus* (left) and *P. aeruginosa* (right) grown in TSB medium with different carbon sources. SEM from a minimum of four biological replicates each consisting of three technical replicates. * indicates a significantly greater concentration of ATP (two-tailed t-test, $p \leq 0.030$). (**D**) Absolute growth of *S. aureus* (left) and *P. aeruginosa* (right) grown in TSB medium with different carbon sources. Absolute growth calculated using data shown in panels B and C. SEM from a minimum of three biological replicates. * indicates a significant difference in absolute growth between both species (two-tailed t-test, $p \leq 0.046$). (**E**) Ratio of absolute growth between *P. aeruginosa* and *S. aureus*. Data from panel D. * indicates a significant difference as determined using data in panel D.

The online version of this article includes the following figure supplement(s) for figure 1:

**Figure supplement 1.** Determining growth rate.

## Results

### Carbon source defined growth environments affects the final ratio of *P. aeruginosa* to *S. aureus*

First, we used a microplate reader to produce growth curves that allowed us to separately quantify growth rate of *P. aeruginosa* and *S. aureus* in the presence of 12 different carbon sources. We chose to use different carbon sources to perturb growth and metabolism in *P. aeruginosa* and *S. aureus* as each species has different carbon source preferences. For example, glucose is a preferred carbon source for *S. aureus*, which results in fast growth and high metabolism (*Halsey et al., 2017*). Alternatively, *P. aeruginosa* prefers organic acids or amino acids and grows slower in glucose coinciding with a reduction in metabolism (*Rojo, 2010*). We observed that in almost all cases, with the exception of α-ketoglutarate, lactic acid, glycerol, and succinate, *S. aureus* had a significantly faster maximum growth rate as compared to *P. aeruginosa* (*Figure 1B*). Next, we quantified the concentration of ATP produced during mid-log phase and in the presence of different carbon sources using a

bioluminescent assay. We note that ATP has been previously shown to be a strong correlate of other measures of metabolism, including oxygen consumption rate and the NAD+/NADH ratio (*Lopatkin et al., 2019*). We observed a range of concentrations of ATP (*Figure 1C*). For example, when grown in medium containing glucose, glycerol, lactic acid, mannose, ribose, and sucrose, *S. aureus* had a significantly higher concentration of ATP compared to *P. aeruginosa*. *P. aeruginosa* had a significantly higher concentration of ATP when grown in the presence of acetate and α-ketoglutarate. For carbon sources of pyruvate, galactose, succinate, lactose, and sorbitol there was not a significant difference in the amount of ATP between the species.

To examine the combined influence of both growth and metabolism on the co-existence of both species we developed a metric called absolute growth, which we define as the ratio of [ATP] (µM) to growth rate (/hr). Examining the combined influence of both growth and metabolism is important as together they may impact the co-existence of both bacteria. For example, consider a carbon source that provides high metabolism and slow growth to *S. aureus*. Here, *S. aureus* may be buffered against the effects of HQNO, pyoverdine and pyochelin, but may be outcompeted by *P. aeruginosa* owing to its slow growth. Alternatively, if a carbon source provides both fast metabolism and growth to *S. aureus*, it is likely to outcompete *P. aeruginosa*. Importantly, previous work has demonstrated that growth and metabolism are not linearly correlated and that nutrient availability can alter the relationship between these two variables (*Lopatkin et al., 2019*; *Manzoni et al., 2012*; *Lahtvee et al., 2011*). We found that, when provided with sucrose or mannose in the growth medium, *S. aureus* had a significantly greater absolute growth relative to *P. aeruginosa* (*Figure 1D*). Conversely, when acetate, α-ketoglutarate, ribose, or pyruvate were provided in the growth medium, *P. aeruginosa* had a significantly greater absolute growth. Providing galactose, glucose, glycerol, lactic acid, lactose, succinate, or sorbitol did not result in a statistically significant difference in absolute growth between both species.

To directly compare absolute growth for a given carbon source and between both species, we determined the ratio of absolute growth of *P. aeruginosa* to *S. aureus*. Ratios >1 indicate that *P. aeruginosa* has a greater absolute growth, whereas ratios <1 indicate that *S. aureus* has a greater absolute growth. We observed a range of absolute growth ratios (*Figure 1E*). *S. aureus* had significantly higher absolute growth when sucrose or mannose was supplied in the growth medium. Alternatively, *P. aeruginosa* had significantly higher absolute growth when ribose, acetate, α-ketoglutarate, or pyruvate was included in the growth medium. Finally, we observed no significant difference in this ratio when galactose, glucose, glycerol, lactic acid, lactose, succinate, or sorbitol was included in the growth medium. Overall, changes in carbon source provided in the growth medium alter both growth rate and metabolism leading to differences in absolute growth between both bacterial species.

## Differences in absolute growth affect the final population composition of *S. aureus* and *P. aeruginosa*

To investigate how absolute growth impacts the co-existence of *S. aureus* and *P. aeruginosa*, we chose seven representative carbon sources that spanned the range of observed absolute growth values. Thus, we chose carbon sources that favored *S. aureus* (sucrose), *P. aeruginosa* (pyruvate, ribose, α-ketoglutarate) or neither (lactic acid, glucose, succinate). We initiated a co-culture of both bacterial species with equal initial densities and grew them for 24 hours without shaking at 37 °C. Final bacterial density in colony forming units CFU/mL was measured using selective plating; cetrimide plates were used to select for *P. aeruginosa* and mannitol salt agar plates were used to select for *S. aureus*. We then determined the final density ratio of *P. aeruginosa* to *S. aureus*; values greater than one would indicate dominance of *P. aeruginosa* in the growth environment. Conversely, values less than one would indicate dominance of *S. aureus* in the growth environment. We observed that differences in the absolute growth ratio determined the final population composition; as the ratio of absolute growth increased, thus increasingly favoring *P. aeruginosa*, the final ratio of *P. aeruginosa* to *S. aureus* significantly increased (p<0.0001, $R^2$=0.97, *Figure 2A*). This finding was consistent when we used a weighted least squares regression (WLS), which considers error along the y-axis when performing a regression (p<0.0001). We also found significant differences amongst the final density ratio (p=0.0291 (Kruskal-Wallis, Shapiro-Wilk for normality, p<0.0001)). We did not find a significant relationship between the final density ratio of *P. aeruginosa* to *S. aureus* and [ATP] or growth rate of either species when plotted independently (*Figure 2—figure supplement 1*). Similarly, we did not find a significant relationship between the final density ratio of *P. aeruginosa* to *S. aureus* and the ratio of growth rates

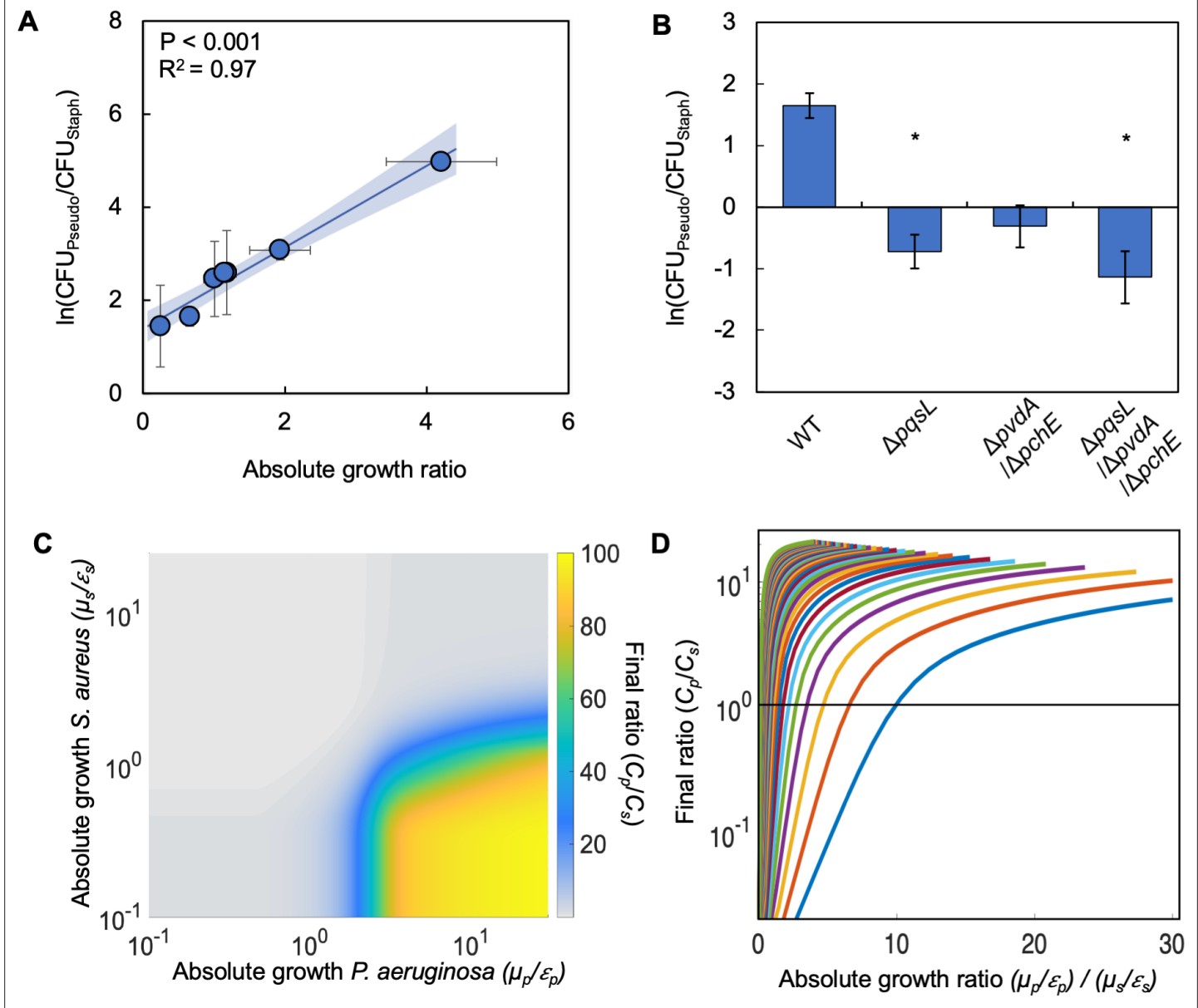

**Figure 2.** Differences in absolute growth determine the final densities of *S. aureus* and *P. aeruginosa* in co-culture. (**A**) Final density ratio of *P. aeruginosa* to *S. aureus* after 24 hr of growth in co-culture with different carbon sources affording different absolute growth ratios. Standard error of the mean (SEM) from a minimum of four biological replicates. $R^2$ and P values shown on plot are from a linear regression. Weighted least squares regression: $R^2$=0.99, p<0.0001. Kruskal-Wallis, p=0.0151 (Shapiro-Wilk<0.0001). For panels A and B, final cell density in CFU for both strains shown in *Figure 2—figure supplement 1*. Linear regressions between the final density ratio and [ATP], maximum growth rates, the ratio of [ATP], and the ratio of growth rates shown in *Figure 2—figure supplement 1*. For panels A and B, exact number of biological replicates and all P values shown in *Supplementary file 2*. Shaded region indicates 95% confidence interval. (**B**) Final density ratio of *P. aeruginosa* knockout strains to *S. aureus* after 24 hr of growth in co-culture. Carbon source included in the growth medium was glucose. SEM from five biological replicates. * indicates significantly different from the wildtype final density ratio (Dunn test with wildtype ratio as control group; p≤0.0309; Kruskal-Wallis; p=0.0051; Shapiro-Wilk<0.001). (**C**) Heat map showing the effect of absolute growth ($\mu/\varepsilon$) of *P. aeruginosa* ($C_p$) and *S. aureus* ($C_s$) on the final ratio of the strains. For these simulations, growth rate ($\mu$) was held constant while metabolism ($\varepsilon$) was varied. For simulations where growth rate is varied and metabolism is held constant, see *Figure 2—figure supplement 2*. For panels C and D, simulations were performed using *Equations 1–3*. Total simulation time = 24 hr. Parameters in *Supplementary file 2*. Model description and development in *Methods*. Sensitivity analysis in *Supplementary file 2*. (**D**) Representative simulations showing the relationship between the ratio of absolute growth and the final density ratio of *P. aeruginosa* to *S. aureus*. For these simulations, growth rate ($\mu$) and metabolism ($\varepsilon$) for *P. aeruginosa* were varied while they were fixed for *S. aureus*. Each colored line presents a combination of $\mu$ and $\varepsilon$ for *P. aeruginosa*. Simulations using fixed values for *P. aeruginosa* and varied values for *S. aureus* shown in *Figure 2—figure supplement 2*.

The online version of this article includes the following figure supplement(s) for figure 2:

*Figure 2 continued on next page*

*Figure 2 continued*

**Figure supplement 1.** Raw density (CFU/mL) of *P. aeruginosa* and *S. aureus* in co-culture in TSB medium.

**Figure supplement 2.** Additional simulations and sensitivity analysis using our mathematical model (*Equations 1–3*).

(growth rate *P. aeruginosa*/growth rate *S. aureus*, *Figure 2—figure supplement 1*). However, we did find a significant (p=0.0076), albeit weaker ($R^2$=0.79), relationship between the final density ratio of *P. aeruginosa* to *S. aureus* and the ratio of [ATP] ([ATP] *P. aeruginosa*/[ATP] *S. aureus*, *Figure 2—figure supplement 1*). Finally, and consistent with our findings above, we observed that when *S. aureus* and *P. aeruginosa* were co-cultured in a more anaerobic environment (with mineral oil over top of the growth medium), increasing the absolute growth ratio increased the dominance of *P. aeruginosa* in the community (*Figure 2—figure supplement 1*).

To provide evidence that the relationship between the final ratio of *P. aeruginosa* to *S. aureus* and absolute growth was attributed to the production of virulence factors by *P. aeruginosa* that perturb metabolism and growth, we acquired knockout strains which lack the ability to synthesize HQNO (Δ*pqsL*), pyoverdine and pyochelin (Δ*pvdA*/Δ*pchE*), and all three virulence factors (Δ*pqsL*/Δ*pvdA*/Δ*pchE*). When knockout strains lacking *pqsL* were co-cultured with *S. aureus* in the presence of glucose as the carbon source, there was a significant decrease (p≤0.0309 [Dunn's test with control for joint ranks, Shapiro-Wilk; p<0.0001]) in the final density ratio as compared to co-culture with wildtype *P. aeruginosa* (*Figure 2B*). The decrease in the final density ratio was owing to a relative increase in the density of *S. aureus*. We did not find a significant reduction in the growth rate of the knockout strains relative to the wildtype strain in all carbon sources tested in *Figure 2A* (*Figure 2—figure supplement 1*). Thus, the reduction in the final density ratio in these experiments is not due to a decrease in growth rate owing to the removal of *pqsL* or additional virulence factors. Overall, this indicated that the production of HQNO, but not necessarily pyoverdine or pyochelin, plays a pivotal role in determining the interactions between *S. aureus* and *P. aeruginosa* in our experimental setup.

Toward understanding why differences in absolute growth affected the co-existence of *P. aeruginosa* to *S. aureus*, we created a simple mathematical model consisting of three ordinary differential equations. The model considers the production of virulence factors (*vir*) by *P. aeruginosa* that reduce growth and metabolism, and the growth and death of both species independently (see *Methods* for equations, model development and parameter estimation; see *Supplementary file 2* for parameters). Production of virulence factors is modeled as a modified Hill Equation that is scaled based on the density of *P. aeruginosa*. The growth of both bacterial species follows a modified logistic growth equation where growth is a product of basal growth rate, *μ*, and metabolism, *ε*. In general, *ε* approximates a maintenance coefficient, which refers to the amount of ATP that is not directly involved in generating biomass. The equation governing the growth of *S. aureus* is modified to account for the effect of virulence factors produced by *P. aeruginosa*. Here, *ε* is scaled by the amount of virulence factors (*vir*) in the system; higher concentrations of virulence factors reduce the value of *ε*. This serves to reduce the entire growth term, thus reducing the rate at which *S. aureus* grows. Consistent with our experimental definition, absolute growth for each species can be determined by calculating *ε/μ*. Our model predicts that, over a wide range of value of *μ* and *ε*, increasing the ratio of absolute growth leads to an increase in the density of *P. aeruginosa* relative to *S. aureus* (*Figure 2C–D*, *Figure 2—figure supplement 2*).

## The ratio of absolute growth predicts population composition in additional growth media

To determine if the relationship between the absolute growth ratio and the final density ratio held true in additional growth medium, we co-cultured *P. aeruginosa* and *S. aureus* in modified synthetic minimal medium (AMM). Unlike other minimal media, such as MOPS or M9, AMM allows the growth of *S. aureus* (*Rudin et al., 1974*). For consistency, we used the same seven carbon sources used to describe the relationship between the final density ratio and the absolute growth ratio in TSB. Similar to our results in TSB medium, we found that changing the carbon source allowed for a range of growth rates (*Figure 3A*) and ATP concentrations (*Figure 3B*), only some of which were significantly different between both bacteria. We found that only some of the absolute growth values (glucose, ribose and sucrose) were significantly different between each species (*Figure 3C–D*). Importantly, we found a strong linear relationship between the absolute growth ratio and the final population composition; as

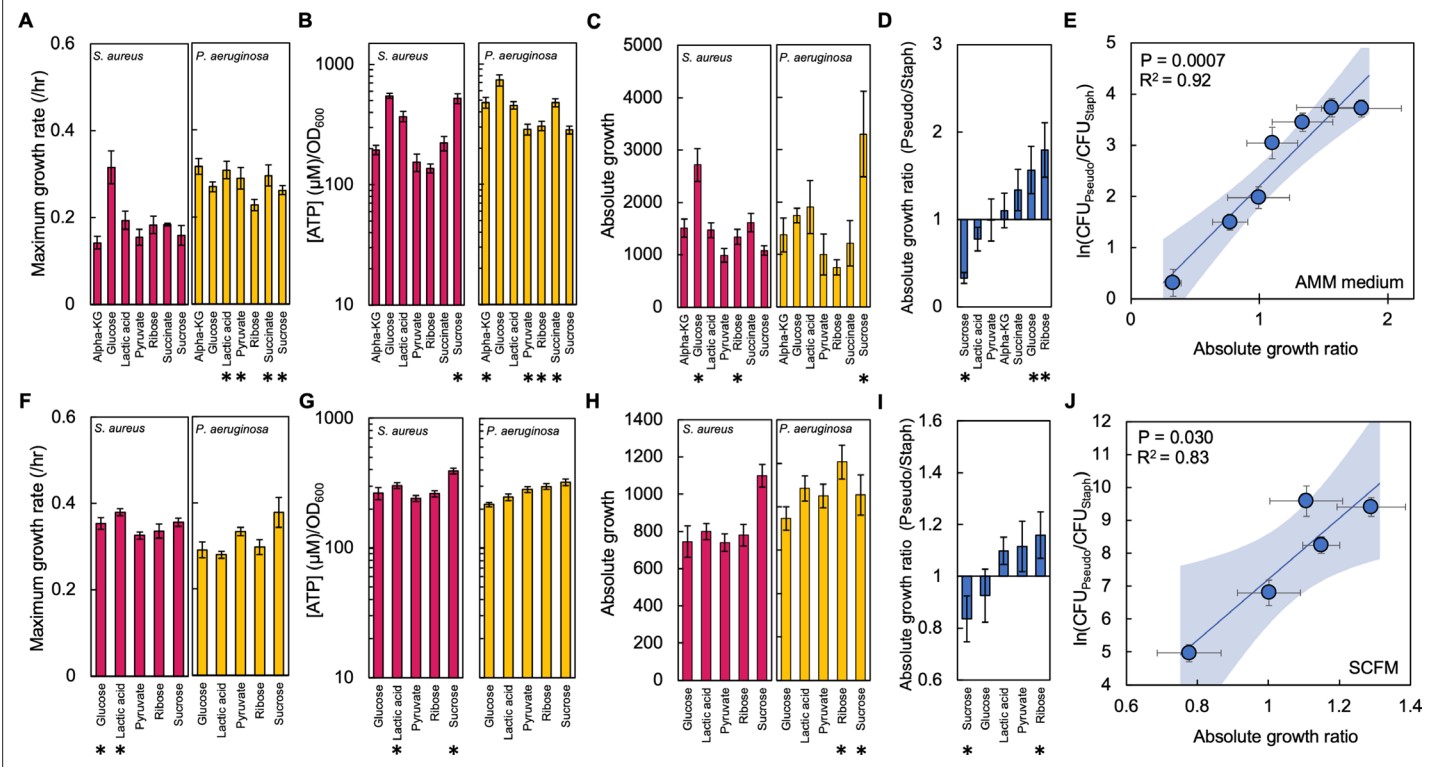

**Figure 3.** The absolute growth ratio predicts community composition in additional growth media. (**A**) Maximum growth rate of *S. aureus* (left) and *P. aeruginosa* (right) when grown in AMM medium with different carbon sources. Growth curves in *Figure 3—figure supplement 1*. Standard error of the mean (SEM) from a minimum of eight biological replicates. * indicates a significantly greater maximum growth rate (two-tailed t-test, $P \leq 0.0033$). For all panels in this figure, the exact number of biological replicates and all p values are in *Supplementary file 4*. (**B**) The concentration of ATP (µM) produced by *S. aureus* (left) and *P. aeruginosa* (right) grown in AMM medium with different carbon sources. Concentration of ATP determined using a standard curve. SEM from six biological replicates each consisting of two technical replicates. * indicates a significantly greater concentration of ATP (two-tailed t-test, $P \leq 0.011$). (**C**) Absolute growth of *S. aureus* (left) and *P. aeruginosa* (right) grown in AMM medium with different carbon sources. Absolute growth calculated using data shown in panels A and B. * indicates a significant difference in absolute growth between both species (two-tailed t-test, $p \leq 0.03$). SEM from a minimum of six biological replicates. (**D**) Ratio of absolute growth between *P. aeruginosa* and *S. aureus* when grown in AMM medium. Data from panel C. * indicates a significant difference as determined using data in panel C. (**E**) Final density ratio of *P. aeruginosa* to *S. aureus* after 24 hr of growth in co-culture in AMM medium with different carbon sources affording different absolute growth ratios. SEM from a minimum of five biological replicates. $R^2$ and p values shown on plot are from a linear regression. Weighted least squares regression: $R^2=0.93$, $p=0.0005$. Kruskal-Wallis, $p<0.0001$ (Shapiro-Wilk=0.0001). Final cell density in CFU/mL in *Figure 3—figure supplement 2*. Linear regressions between the final density ratio and [ATP], maximum growth rates, the ratio of [ATP], and the ratio of growth rates shown in *Figure 3—figure supplement 2*. Shaded region indicates 95% confidence interval. (**F**) Maximum growth rate of *S. aureus* (left) and *P. aeruginosa* (right) when grown in SCFM with different carbon sources. Growth curves in *Figure 3—figure supplement 1*. SEM from a minimum of five biological replicates. * indicates a significantly greater maximum growth rate (two-tailed t-test, $p \leq 0.035$). (**G**) The concentration of ATP (µM) produced by *S. aureus* (left) and *P. aeruginosa* (right) grown in SCFM with different carbon sources. SEM from five biological replicates each consisting of two technical replicates. * indicates a significantly greater concentration of ATP (two-tailed t-test, $p \leq 0.04$). (**H**) Absolute growth of *S. aureus* (left) and *P. aeruginosa* (right) grown in SCFM with different carbon sources. Absolute growth calculated using data shown in panels F and G. * indicates a significant difference in absolute growth between both species (two-tailed t-test, $p \leq 0.04$). (**I**) Ratio of absolute growth between *P. aeruginosa* and *S. aureus* when grown in SCFM. Data from panel H. * indicates a significant difference as determined using data in panel H. (**J**) Final density ratio of *P. aeruginosa* to *S. aureus* after 24 hr of growth in co-culture in SCFM with different carbon sources affording different absolute growth ratios. SEM from a minimum of four biological replicates. $R^2$ and p values shown on plot are from a linear regression. Weighted least squares regression: $R^2=0.91$, $p=0.0128$. Kruskal-Wallis, $p<0.0001$ (Shapiro-Wilk <0.0001). Final cell density in CFU/mL shown in *Figure 3—figure supplement 2*. Linear regressions between the final density ratio and [ATP], maximum growth rates, the ratio of [ATP], and the ratio of growth rates shown in *Figure 2*. Shaded region indicates 95% confidence interval.

The online version of this article includes the following figure supplement(s) for figure 3:

**Figure supplement 1.** Raw growth curves of bacteria grown in either (**A**) AMM medium or (**B**) SCFM.

**Figure supplement 2.** Raw density (CFU/mL) of *P. aeruginosa* and *S. aureus* in co-culture in AMM medium and SCFM.

the absolute growth ratio increased, so did the final density ratio (p=0.0007, R²=0.92, **Figure 3E**). This finding was consistent using a WLS regression (p=0.0005, R²=0.93) and there were significant differences amongst the final density ratios (p<0.0001, Kruskal-Wallis; Shapiro-Wilk, p<0.0001).

Next, we co-cultured both *P. aeruginosa* and *S. aureus* in synthetic cystic fibrosis medium (SCFM), which was developed to mimic the composition of sputum isolated from individuals with cystic fibrosis (**Palmer et al., 2007**). As above, we measured growth rate and the concentration of ATP in SCFM containing the seven representative carbon sources used in TSB. However, reliable quantification of growth rate for *P. aeruginosa* grown in SCFM with either succinate or α-ketoglutarate could not be achieved in our experimental setup. Thus, we focused our analysis on the five remaining carbon sources (glucose, sucrose, ribose, pyruvate, lactic acid). We found significant differences between both species in terms of their growth rate (**Figure 3F**) and the concentration of ATP (**Figure 3G**). While only the use of ribose or sucrose as a carbon source led to a statistically significant difference in absolute growth between both species (**Figure 3H–I**), we continued to find a significant linear relationship between the final density ratio and the ratio of absolute growth (p=0.030, R²=0.83, **Figure 3J**). This relationship remained consistent when tested using a WLS regression (p=0.0128, R²=0.91) and we found significant differences amongst the final densities of *S. aureus* and *P. aeruginosa* (p<0.0001, Kruskal-Wallis; p<0.0001, Shapiro-Wilk). Similar to our work in TSB medium, we did not find significant relationships between the final density ratio and [ATP] or growth rate of either species when plotted independently or as ratios for both modified AMM medium and SCFM (**Figure 3—figure supplement 2**). Overall, the relationship between the final density ratio and the absolute growth ratio was consistent in three different growth media with different compositions.

## The ability of *S. aureus* to metabolically buffer against virulence factors is dependent upon absolute growth and the initial population composition

To gain intuition into these findings, we sought to understand how altering the initial ratio of *P. aeruginosa* and *S. aureus* would affect their final ratio after 24 hr of co-culture. Previous work has indicated that the initial population composition can impact the long-term co-existence and composition of microbial communities (**Wright et al., 2021**). Moreover, the ability to predict the formation of microbial communities based on early conditions can allow the prediction of long-term composition of microbial populations (**Friedman et al., 2017**; **Meroz et al., 2021**), which has implications in the rationale design of such communities and in infectious disease.

First, we used our mathematical model to simulate the effect of altering the initial ratio of both species. Our model predicts that, for a given absolute growth ratio, as the initial fraction of *P. aeruginosa* increases, the final population density of *P. aeruginosa* also increases (**Figure 4A**). Our model also predicts that, for a given initial density of *P. aeruginosa*, increasing the absolute growth ratio increases the final density of *P. aeruginosa*. Interestingly, for a given absolute growth ratio, our model predicts an initial fraction of *P. aeruginosa* that serves as a tipping point; if the initial fraction of *P. aeruginosa* exceeds this tipping point, *P. aeruginosa* will dominate the final population (**Figure 4B**). Otherwise, if the initial fraction of *P. aeruginosa* is below this tipping point, *S. aureus* will dominate the final population. The initial fraction of *P. aeruginosa* that leads to this tipping point is determined by the absolute growth ratio; as this ratio increases, the initial fraction of *P. aeruginosa* at the tipping point decreases. In other words, as the absolute growth ratio increases, *P. aeruginosa* will dominate the final population when initiated at lower initial fractions. We note that our model predicts that the amount of *vir* synthesized by *P. aeruginosa* increases based on both increasing initial density and an increasing absolute growth ratio (**Figure 4A**, inset).

Our model predicts that the final population composition is largely owing to interactions between initial density, the synthesis of virulence factors that affect growth and metabolism in *S. aureus,* and the ratio of absolute growth. For a given absolute growth ratio (e.g. 1), if the initial fraction of *S. aureus* ($C_s$ = 0.9) is much greater than *P. aeruginosa* ($C_p$ = 0.1), the amount of virulence factors produced by *P. aeruginosa* is insufficient to reduce the growth of *S. aureus* (**Figure 4C**, top left panel). Coupled with a higher initial starting density, this allows *S. aureus* to outcompete *P. aeruginosa*; *S. aureus* dominates the final population. As the initial fraction of *P. aeruginosa* is increased, additional virulence factors are produced. Once *P. aeruginosa* ($C_p$ = 0.5) reaches an initial fraction that is sufficiently high to produce enough virulence factors to reduce the growth of *S. aureus* ($C_s$ = 0.5), *P. aeruginosa* dominates the

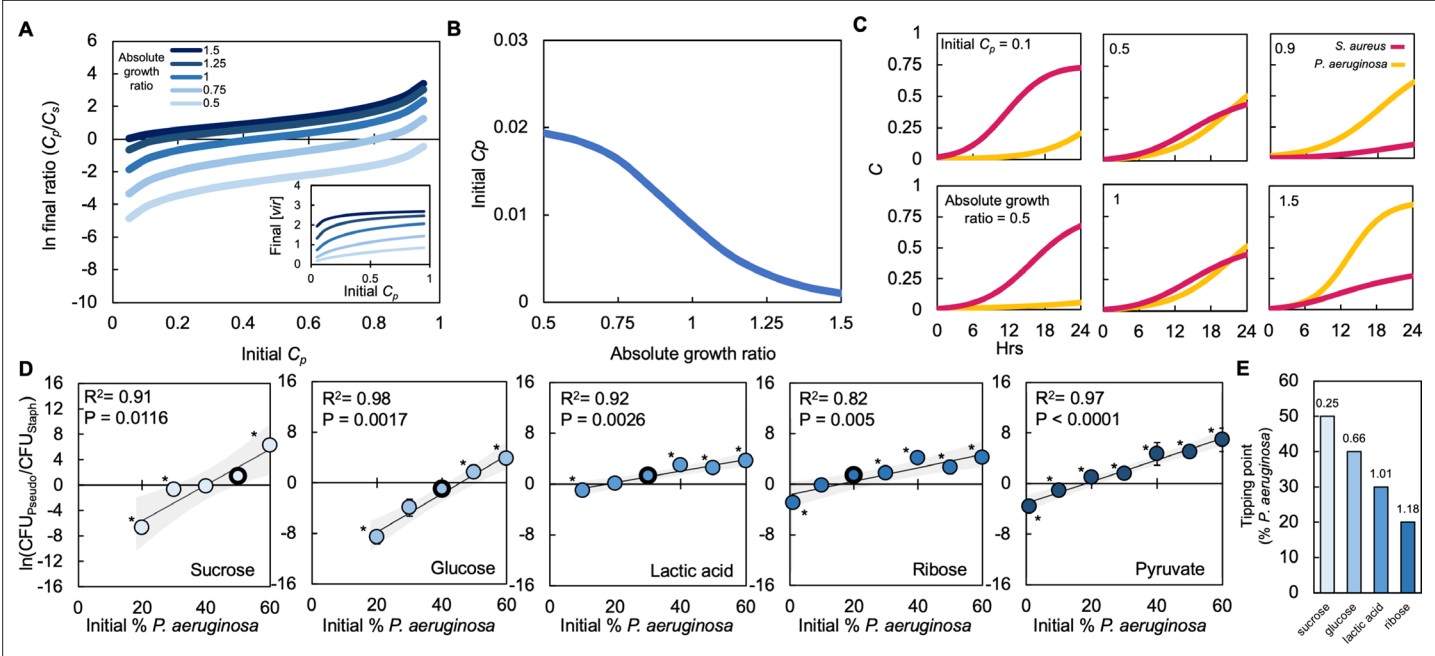

**Figure 4.** Absolute growth determines tipping point of co-existence of *P. aeruginosa* and *S. aureus* in co-culture. (**A**) Simulations showing the effect of changing the initial fraction of *P. aeruginosa* ($C_p$) and *S. aureus* ($C_s$) on the final density ratio. Each colored line represents simulations performed with different absolute growth values (as indicated in the figure legend). For panels A-C, simulations performed using *Equations 1–3*. Total simulation time = 24 hours. Parameters in *Supplementary file 2*. Model description and development in *Methods*. Inset: Simulations showing the final concentration of [*vir*] produced by *P. aeruginosa* as a function of initial density and at different absolute growth ratios. (**B**) Tipping point. Simulations showing the lowest initial fraction of *P. aeruginosa* ($C_p$) that results in a final density ($C_p/C_s$) of 1 after 24 hr as a function of the absolute growth ratio. In these simulations, metabolism ($\varepsilon$) for *P. aeruginosa* was varied but metabolism for *S. aureus* and maximum growth rates ($\mu$) of both species were held constants. For simulations showing the effect of varying $\varepsilon$ see *Figure 4—figure supplement 1*. (**C**) Simulations showing the temporal changes in the density of *P. aeruginosa* ($C_p$) and *S. aureus* ($C_s$). Top panels show the effect of increasing the relative initial density of $C_p$ at the same absolute growth ratio. Bottom panels show the effect of increasing the absolute growth ratio while maintaining the same initial population densities ($C_p = 0.5$, $C_s = 0.5$). (**D**) Experimental data: The final density of a co-cultured population grown in TSB medium with different carbon sources plotted as a function of the initial percentage of *P. aeruginosa*. Standard error of the mean (SEM) from a minimum of three biological replicates. $R^2$ and p values shown on plot are from a linear regression. * indicates a significant difference in the final density of *P. aeruginosa* and *S. aureus* (Mann-Whitney, p≤0.0463) (Shapiro-Wilk for normality ≤0.0001). When all final density ratios are compared within each carbon source p≤0.0029, Kruskal-Wallis. The exact number of biological replicates and all P values are in *Supplementary file 5*. Final cell density in CFU/mL shown in *Figure 4—figure supplement 1*. Data points outlined in black indicate the tipping point; the lowest initial percentage of *P. aeruginosa* examined that led to a significantly greater amount of *P. aeruginosa* after 24 hr of growth. Shaded region indicates 95% confidence interval. (**E**) Tipping point of the population defined as the first initial percentage of *P. aeruginosa* measured where there was not a significant difference in the final density of *P. aeruginosa* and *S. aureus*. Data from panel D. Number above each bar indicates absolute growth ratio calculated using data in *Figure 1E*. We did not find the tipping point for pyruvate and thus it is not included on this plot.

The online version of this article includes the following figure supplement(s) for figure 4:

**Figure supplement 1.** Raw bacteria density (CFU/mL) for co-cultures that were grown in TSB medium with different carbon sources.

population after 24 hr of growth (*Figure 4C*, top center panel). Finally, if the initial fraction of *P. aeruginosa* ($C_p = 0.9$) is much greater than *S. aureus* ($C_s = 0.1$) the growth of *S. aureus* is sufficiently hampered such that *S. aureus* does not grow appreciably and *P. aeruginosa* dominates the population (*Figure 4C*, top right panel).

Our model predicts that, for a given initial density ratio ($C_p = C_s = 0.5$), increasing the absolute growth ratio benefits the growth of *P. aeruginosa*. If the absolute growth ratio is sufficiently small (0.5), *P. aeruginosa* is unable to synthesize a high amount of virulence factors and grows slower than *S. aureus*. Slow growth of *P. aeruginosa* coupled with a low amount of virulence factors synthesized allows *S. aureus* to dominate the population (*Figure 4C*, bottom left panel). As the absolute growth ratio increases, it serves to benefit *P. aeruginosa*. At an intermediate absolute growth ratio (1), the amount of virulence factors produced is sufficient to reduce the growth rate of *S. aureus* while enhancing the growth rate of *P. aeruginosa*. Combined, this allows *P. aeruginosa* to dominate the population

(*Figure 4C*, bottom center panel). Finally, when the absolute growth ratio is large (1.5), a high amount of virulence factors is produced, which substantially reduces the growth of *S. aureus*. The increase in absolute growth ratio serves to further increase the growth rate of *P. aeruginosa*. Together, this allows *P. aeruginosa* to significantly outcompete *S. aureus* (*Figure 4C*, bottom right panel).

To confirm these modeling predictions, we grew *P. aeruginosa* and *S. aureus* in co-culture as described above. We varied the initial percentage between both species and used five of the same carbon sources as used previously (*Figure 2A*). We observed that our experimental results match the qualitative predictions made by our model. First, we observed that, for all absolute growth ratios tested, as the initial percentage of *P. aeruginosa* increased, the final population density was increasingly dominated by *P. aeruginosa*, which resulted in a greater final density ratio (*Figure 4D*). If the initial percentage of *P. aeruginosa* was sufficiently small, *S. aureus* could dominate the population, leading to a smaller final density ratio. Moreover, as predicted by our model, we found that, as the ratio of absolute growth increased, the greatest initial percentage of *P. aeruginosa* measured where neither *P. aeruginosa* nor *S. aureus* dominated the population after 24 hr (or the tipping point) decreased (*Figure 4D–E*). For example, when grown in the presence of sucrose, of which *S. aureus* has a higher absolute growth, the greatest initial percentage of *P. aeruginosa* required such that its final density was not different than *S. aureus* after 24 hr of growth was 50% (*Figure 4E*). Conversely, when grown in ribose, where *P. aeruginosa* has a higher absolute growth, the greatest initial percentage required for the density of *P. aeruginosa* to be no different than *S. aureus* after 24 hr of growth was 20%, considerably smaller than sucrose (*Figure 4E*). Overall, both our model and experimental analysis indicate that interactions between initial population composition, the synthesis of virulence factors and absolute growth determined population composition after 24 hr of growth.

## Spatial overlap and absolute growth affect co-existence of *P. aeruginosa* and *S. aureus*

Previous work has shown that the relative distribution of bacteria that interact through small diffusible molecules or proteins, such as HQNO, can influence population dynamics (*Darch et al., 2018*). In a stationary environment, if bacteria are positioned close together, the effective concentration of a small diffusible molecule sensed by the bacteria will be higher than if bacteria are positioned farther apart. Conversely, if the distribution of bacteria in the environment is homogenous, such as in a continuously shaken condition, the concentration of small diffusible molecules sensed will be roughly equal across the population. Furthermore, it has been previously shown that periodically disturbing the spatial structure of a bacterial population using a physical force not only serves to disrupt the distribution of bacteria but can also disrupt access to small diffusible molecules (*Quinn et al., 2021*; *Wilson et al., 2017*; *Barraza et al., 2023*). Given that diffusible virulence factors HQNO, pyoverdine and pyochelin influence the interactions between *P. aeruginosa* and *S. aureus*, we sought to determine how the distribution of bacteria, coupled with differences in absolute growth, determined the final population composition.

To understand the relationship between the distribution of bacteria, differences in absolute growth, and their impact on population composition, we modified our series of ODEs to include a spatial overlap term, $\delta$, which scales the amount of virulence factor (*vir*) sensed by *S. aureus*. High values of $\delta$ indicate a high degree of spatial overlap between the two populations; decreasing $\delta$ indicates reduced overlap, which diminishes the amount of virulence factors sensed by *S. aureus*. Experimentally, estimating $\delta$ would require the tracking of the positioning of multiple bacteria over both space and time whereupon a spatial overlap value could be extracted. Moreover, any movement of the plate for examination via microscopy would serve to disrupt the distribution of the cells, thus obscuring the results. Due to these challenges, we took a modeling approach to estimate trends in $\delta$.

We used an agent-based model (ABM, see *Methods*) to estimate the extent of overlap of the two bacterial species. The simplified ABM consists of two populations of agents that are randomly placed in a world. The agents grow and die according to logistic growth. The agents also undergo a displacement event where the spatial positioning of the agents is perturbed at a given distance (called distance traveled). Increasing the number of displacement events per hour increases the degree to which the spatial positions of the agents are perturbed each hour of the simulation. In the absence of any disturbance events, the populations of bacteria do not move, which represents a stationary condition. When the number of disturbance events is very high, the population becomes well-mixed,

which approximates a continuously shaken culture. An intermediate value of disturbance events would represent a population whose spatial overlap is perturbed periodically. After every hour of in silico simulation time using the ABM, we measure the extent of spatial overlap of the population frequency of cohabited patches minus the expected frequency of cohabited patches (frequency of patches inhabited by *P. aeruginosa* agents x frequency of patches inhabited by *S. aureus* agents). We call this metric 'cohabitation'. A negative cohabitation value means that the populations spatially overlap less than expected based on random chance; a positive cohabitation value means that the populations overlap more than expected. The cohabitation variable is calculated every hour, which is then averaged across the 24 hr in silico time period.

We observed that, over a wide range of initial population densities (*Figure 5A*) and distances traveled (*Figure 5—figure supplement 1*), cohabitation followed a biphasic trend; this value first decreases with increasing disturbance events whereupon it begins to increase with increasing disturbance events. When the number of disturbance events is low (0 /hr) or very high (50 /hr), cohabitation is high. Toward the former, this is owing to the slow relative breakdown of their initial high degree of overlap. Toward the latter, this is due to the well-mixed nature of the population where there is significant overlap between the agents. In between these two extremes the value of cohabitation is lower. This is owing to disturbance spreading-out populations and distributing cells to unoccupied areas to grow faster in the relative absence of competitors. As disturbance is infrequent, each agent spends more time in an unoccupied area over the course of the simulation, which reduces the value of cohabitation.

Using the trends in cohabitation predicted by our ABM, we simulated the effect of changing $\delta$ and absolute growth of *P. aeruginosa* on the final density of both populations. For simplicity, we kept the absolute growth of *S. aureus* constant. We observed that at low values of absolute growth for *P. aeruginosa*, $\delta$ has little effect as *S. aureus* dominates the culture (*Figure 5B*). However, at higher values of absolute growth for *P. aeruginosa*, the value of $\delta$ affects the final population composition. If $\delta$ is sufficiently large, *P. aeruginosa* dominates the population. Otherwise, if $\delta$ is sufficiently small, such as observed at environments that are disturbed at intermediate frequencies, *S. aureus* dominates the population.

To understand these modeling predictions, we simulated temporal changes in cell density for both species independently and at three values of $\delta$ (*Figure 5C–D*). When $\delta$ is small (0.1), which represents a co-culture disturbed at an intermediate frequency (*Figure 5A*), *S. aureus* grows quickly reaching a high density after 24 hr. This high growth rate is owing to a reduction in the amount of virulence factors that are sensed by *S. aureus*. At intermediate values of $\delta$ (0.5), which continues to represent a co-culture disturbed at an intermediate frequency (*Figure 5A*), the amount of virulence factors sensed by *S. aureus* increases, which reduces its overall growth rate. However, because the growth rate of *P. aeruginosa* is sufficiently small owing to low absolute growth, *S. aureus* continues to dominate the population after 24 hr of growth. Finally, when $\delta$ is large (1), which represents a co-culture that is undisturbed (stationary) or very frequently disturbed (continuous shaking) the amount of virulence factors sensed by *S. aureus* is high. This significantly reduces its overall growth rate, allowing *P. aeruginosa* to outcompete it. Accordingly, *P. aeruginosa* dominates the population after 24 hr under this condition. Importantly, under each of these simulated conditions the total amount of virulence factor synthesized is not altered ([*vir*]=1.77 μM); only the concentration sensed by *S. aureus* is altered owing to changes in $\delta$.

To test these predictions, we periodically disturbed co-cultured populations of *S. aureus* and *P. aeruginosa* using the linear shaking function of a microplate reader. We previously demonstrated that using this function could alter the spatial distribution of bacteria (*Quinn et al., 2021*; *Wilson et al., 2017*). We grew both bacterial species using glucose as the carbon source as it represented an intermediate ratio of absolute growth. As above, when both bacteria species were initiated at equal percentages, the use of glucose allowed *P. aeruginosa* to dominate the population in the stationary condition. However, as predicted by our model, *S. aureus* dominated the final population composition over a range of intermediate disturbance frequencies (3 /hr-18/hr, *Figure 5E*). At disturbance frequencies of 1 /hr and 20 /hr, there was no significant difference in the density of *P. aeruginosa* and *S. aureus*. Finally, when co-cultures were perturbed continuously or not all (0 /hr), the final density of *P. aeruginosa* was significantly greater than that of *S. aureus*. We note that *S. aureus* was observed to dominate the population at a disturbance frequency of 6 /hr at lower initial dilutions of the population

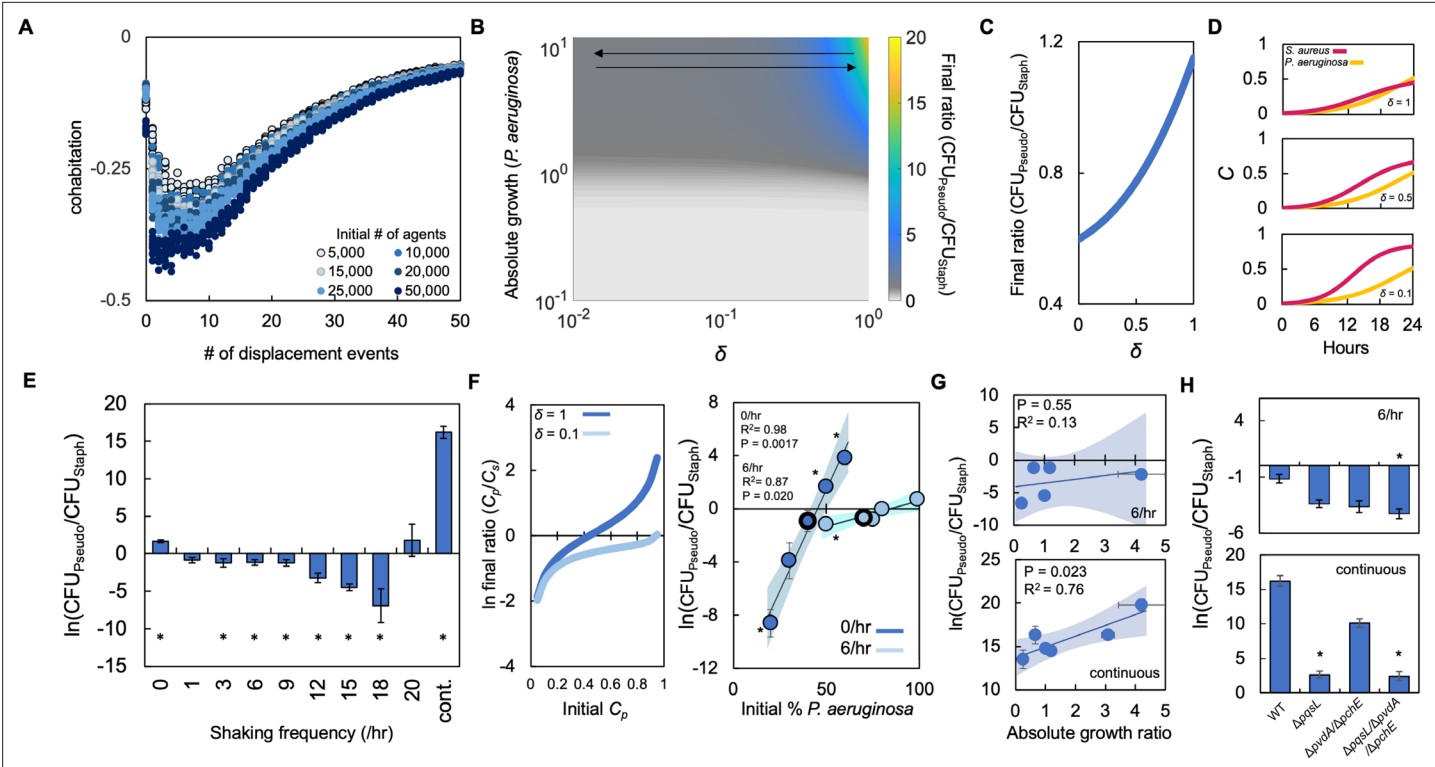

**Figure 5.** Periodically disturbing co-cultured populations of *P. aeruginosa* and *S. aureus* determines their co-existence. (**A**) Simulation using our ABM showing the 'cohabitation' variable as a function of the number of disturbance events. These simulation trends were used to estimate trends in *δ* (*Equations 1–3*). Each simulation was performed 10 times; each data point represents an outcome from a single simulation. See *Methods* for ABM description and parameter justification. Sensitivity analysis showing U-shaped trend over a wide range of parameters displayed in *Figure 5—figure supplement 1*. (**B**) Heat map showing the effect of *δ* and absolute growth of *P. aeruginosa* (fixed absolute growth of *S. aureus*) on the final ratio of *P. aeruginosa* and *S. aureus*. For panels, B-D, simulations performed using *Equations 1–3*. Total simulation time = 24 hr. Parameters in *Supplementary file 2*. Model description and development in *Methods*. (**C**) Simulations showing the effect of changing *δ* on the final ratio of *P. aeruginosa* and *S. aureus* for a fixed ratio of absolute growth (1). (**D**) Simulations showing the temporal changes in the density of *P. aeruginosa* ($C_p$) and *S. aureus* ($C_s$) at different values of *δ* as indicated on each panel. (**E**) Experimental results. The ratio of *P. aeruginosa* to *S. aureus* as a function of disturbance events in TSB medium containing glucose as the carbon source. * indicates a significant difference (p≤0.0202 using a Mann-Whitney, Shapiro-Wilk<0.0001) in the final density of both populations. Kruskal-Wallis for all data points; p<0.0001. Raw density data (in CFU/mL) shown in *Figure 5—figure supplement 2*. Standard error of the mean (SEM) from a minimum of three biological replicates. For panels E, G, and H, the exact number of biological replicates and all P values are in *Supplementary file 6*. (**F**) Left: Simulations (*Equations 1–3*) showing the effect of changing δ on the final population density when the initial density of *P. aeruginosa* ($C_p$) is varied. Total simulation time = 24 hr. Parameters in *Supplementary file 2*. Right: Experimental data showing the final density of a co-cultured population either undisturbed (0 /hr, dark blue) or disturbed at 6 /hr (light blue) grown in TSB medium with different carbon sources plotted as a function of the initial percentage of *P. aeruginosa*. SEM from a minimum of three biological replicates. $R^2$ and p values shown on plot are from a linear regression. * indicates a significant difference in the final density of *P. aeruginosa* and *S. aureus* (Mann-Whitney; p≤0.0202, Shapiro-Wilk; p=0.0190). Kruskal-Wallis for all data points: 0 /hr – p=0.0009; 6 /hr – p=0.115. Final cell density in CFU/mL shown in *Figure 5—figure supplement 2*. The exact number of biological replicates and all p values are in *Supplementary file 6*. Shaded regions indicate 95% confidence interval. (**G**) Experimental results. The ratio of *P. aeruginosa* to *S. aureus* after 24 hr of growth and with 6 disturbance events per hour (top) or in a continuously disturbed environment (bottom) in TSB medium containing different carbon sources. Difference between all data points: 6 /hr: p=0.0023 (Kruskal-Wallis); continuous: p<0.0366, (Kruskal-Wallis). Weighted least squares regression: 6 /hr, $R^2$=0.034 p=0.765; continuous $R^2$=0.72 p=0.0332. Raw cell density data (in CFU/mL) shown in *Figure 5—figure supplement 2*. SEM from a minimum of three biological replicates. Shaded regions indicate 95% confidence interval. (**H**) Experimental Results: The effect of six disturbance events per hour and continuous disturbance on the final density ratio of *P. aeruginosa* knockout strains and *S. aureus* wild type strain. Raw density data (in CFU/mL) shown in *Figure 5—figure supplement 2*. * indicates that the final density ratio was significantly different from co-culture with the wildtype strain (p<0.0129, Dunn test for joint ranks using wildtype as the control group). SEM from a minimum of five biological replicates.

The online version of this article includes the following figure supplement(s) for figure 5:

**Figure supplement 1.** Sensitivity analysis from our agent-based model.

**Figure supplement 2.** Raw bacteria density (CFU/mL) for co-cultures that were disturbed periodically.

**Figure supplement 3.** Periodic, but not continuous, disturbance can reduce dominance of *P. aeruginosa* in SCFM.

(*Figure 5—figure supplement 2*). Moreover, we also found that disturbance at 6 /hr shifted the tipping point of the population to a higher initial percentage of *P. aeruginosa* as compared to the stationary condition (0 /hr, *Figure 5F*). This finding was consistent both in silico (*Figure 5F*, left panel) and in vitro (*Figure 5F*, right panel). Consistent with the results observed in TSB medium, we found that disturbance at 6 /hr in SCFM allowed *S. aureus* to increase in density relative to *P. aeruginosa*. This resulted in a reduction in the final density ratio. Continuous disturbance of co-cultures in SCFM allowed *P. aeruginosa* to dominate once again as evidenced by an increase in the final density ratio (*Figure 5—figure supplement 3*).

Next, we investigated how changing absolute growth through the use of different carbon sources might impact the ability of *S. aureus* to dominate at intermediate, but not high, disturbance frequencies, such as a continuously mixed population. Accordingly, we grew co-cultured populations of *S. aureus* and *P. aeruginosa* in four additional carbon sources that spanned the range of absolute growths and disturbed these populations at a disturbance frequency of 6 /hr or continuously. As predicted by our model, at a disturbance frequency of 6 /hr, *S. aureus* dominated the population after 24 hr of growth regardless of the absolute growth ratio (*Figure 5G*, top panel). Under continuous shaking, we observed that as the ratio of absolute growth increased, the final density ratio increased (*Figure 5G*, bottom panel). We note that we did not find a significant relationship between the final density ratio of *P. aeruginosa* to *S. aureus* (at both 6 /hr and continuous disturbance) and the ratio of growth rates (*P. aeruginosa* growth rate/*S. aureus* growth rate or the ratio of [ATP]) (P. *aeruginosa* [ATP]/*S. aureus* [ATP]).

In both the 6 /hr and continuous disturbance conditions, metabolism altering virulence factors produced by *P. aeruginosa* continues to have a predominant role in mediating the interaction between both bacterial species. When *S. aureus* was co-cultured with knockout strains of *P. aeruginosa* (Δ*pqsL*; Δ*pvdA*/Δ*pchE*; Δ*pqsL*/Δ*pvdA*/Δ*pchE*), growth with Δ*pqsL*/Δ*pvdA*/Δ*pchE* resulted in a significant decrease in the final density ratio when disturbed at 6 /hr (*Figure 5H*, top panel). When disturbed continuously, co-culture with either Δ*pqsL* or Δ*pqsL*/Δ*pvdA*/Δ*pchE* led to a significant decrease in the final density ratio (*Figure 5H*, bottom panel). In both cases, the decrease in the final density ratio was owing to a relative increase in the density of *S. aureus*. Consistent with our findings when co-cultured in the stationary condition (*Figure 2B*), these findings generally implicate the production of HQNO in driving the dynamics between *S. aureus* and *P. aeruginosa*. Overall, our simulations and experiments suggest that access to metabolite altering virulence factors driven by spatial organization and absolute growth can determine the final population composition of *P. aeruginosa* and *S. aureus*.

## Discussion

Through a series of empirical and analytical investigations, we have shown that differences in absolute growth, a combined metric that encompasses the relationship between growth rate and metabolism, can determine the population composition of *S. aureus* and *P. aeruginosa*. In a stationary environment, as the absolute growth ratio increases, the final density ratio of *P. aeruginosa* and *S. aureus* increases. The absolute growth ratio also determines the tipping point of the co-cultured population. Importantly, many virulence factors produced by *P. aeruginosa* (e.g., HQNO, pyocyanin) that impact growth and metabolism in *S. aureus* are regulated by quorum sensing. Quorum sensing involves the production and exchange of small diffusible molecules, called autoinducers (*Waters and Bassler, 2005*). Since each *P. aeruginosa* bacterium produces autoinducers, the concentration of autoinducer increases as a function of cell density. At a sufficiently high concentration, the autoinducer will activate expression of multiple virulence factors in *P. aeruginosa*, which begin to impact growth and metabolism in *S. aureus*. Given this density-dependent activation of virulence factors, it is likely that the tipping point in our systems is being determined by quorum sensing, although we did not explicitly investigate this hypothesis in this manuscript.

While the ability of the absolute growth ratio to predict final population composition is also true in a continually disturbed environment, this relationship falls apart in a periodically disturbed environment. Our model and experiments suggest that this is owing to changes in access to metabolism altering virulence factors produced by *P. aeruginosa*; in stationary or continuously disturbed environments, the spatial overlap of both bacteria is predicted to be high, which facilitates interactions between these virulence factors and *S. aureus*. However, in periodically disturbed environments, the overlap between bacteria is reduced; this limits interactions between *S. aureus* and virulence factors. *S. aureus* then

dominates the population. Absolute growth appears to be the most accurate metric within the scope of our study to quantify the changes in the final ratio between *P. aeruginosa* and *S. aureus*. While we did find a significant relationship between the final ratio of both species and the ratio of [ATP] of both species when grown in the stationary condition and in TSB medium, the relationship was not as strong as the absolute growth ratio ($R^2$=0.97 (absolute growth ratio) vs. 0.79 ([ATP] ratio)). Moreover, we did not find a significant relationship between the final ratio of both species and the ratio of [ATP] when grown in TSB medium in the continuously disturbed condition, and in either AMM medium or SCFM in the stationary condition (*Figure 3—figure supplement 2*).

Given the clinical importance of co-infection with both *P. aeruginosa* and *S. aureus*, multiple previous studies have identified mechanisms of co-existence. Indeed, long-term co-existence of both species can result in physiological changes that reduce their competitive interactions. Strains of *P. aeruginosa* isolated from patients that enter into a mucoid state show reduced production of sidero-phores, pyocyanin, rhamnolipids, and HQNO, which facilitates the survival of *S. aureus* (*Limoli et al., 2017*; *Frydenlund Michelsen et al., 2016*). These strains can also overproduce the polysaccharide alginate, which in itself is sufficient to decrease the production of these virulence factors. Moreover, exogenously supplied alginate can reduce the production of pyoverdine and expression from the PQS quorum sensing system, which is responsible for the production of HQNO (*Price et al., 2020*). Changes in the physiology of *S. aureus* can also facilitate co-existence. Strains of *S. aureus* isolated from patients with cystic fibrosis show multiple changes in the abundance of proteins including super oxide dismutase, the GroEL chaperone protein, and multiple surface-associated proteins (*Treffon et al., 2018*). Interestingly, the majority of proteins that show changes in abundance in *S. aureus* are related to central metabolism, which is consistent with our findings demonstrating that metabolism can influence the co-existence of both species. While it is unclear as to how long-term co-culture would affect the ratio of absolute growth, our findings provide an additional mechanism that can determine the co-existence of these bacterial species.

Previous work has shown that niches in the human body, which can vary in environmental conditions, can promote bacterial growth and persistence (*Fung et al., 2019*). Spatiotemporal differences in nutrient availability within these niches can promote the growth of different bacteria (*Pereira and Berry, 2017*; *Hoffman et al., 2010*). Even within the same niche, such as sputum isolated from individuals with cystic fibrosis, the concentration of critical nutrients, such as amino acids and carbon sources, can differ from host to host (*Palmer et al., 2007*). Variations in nutrient availability within or between niches may alter growth and metabolism of the resident bacteria, and may serve to perturb interactions within polymicrobial communities. In the case of *P. aeruginosa* and *S. aureus*, differences in nutrient availability in sputum (*Palmer et al., 2007*) or the epithelium (*Chen et al., 2018*) may alter growth, metabolism and the interactions between these species. This could alter their co-existence dynamics leading to host and/or niche specific colonization dynamics. We note that our findings may be relevant to infections occurring in both high and low $O_2$ environments. While *P. aeruginosa* is limited in its ability to perform fermentation (*Glasser et al., 2014*), we have provided evidence that the absolute growth ratio can affect community composition in both aerobic (*Figures 2–5*) and more anaerobic environments (*Figure 2—figure supplement 1*). The limited ability of *P. aeruginosa* to grow in anaerobic environments was apparent in SCFM as we could not obtain reliable or robustly quantifiable growth of this bacteria when succinate or α-ketoglutarate was provided as a carbon source. One drawback of our approach in using different carbon sources to manipulate absolute growth is that some carbon sources are osmotically active, whereas others are not, which could have additional physiological effects on the bacteria outside of changing growth and metabolism. Moreover, both species of bacteria have different carbon source preferences; as above *S. aureus* tends to prefer carbon sources such as glucose (*Halsey et al., 2017*), whereas *P. aeruginosa* prefers organic and amino acids (*Rojo, 2010*). Given the carbon source preferences of each species, in complex medium such as TSB, there is the potential that *P. aeruginosa* consumes amino acids prior to consuming the supplied carbon source. This is perhaps less of a concern in AMM medium or SCFM where the concentration of amino acids and additional nutrient components is reduced as compared to TSB medium. Along this line, it is certainly worth investigating how each nutrient component and its ordered utilization by both species contributes to changes in absolute growth. Minor or transient changes in absolute growth owing to preferential nutrient consumption may provide windows of opportunity for one species to increase its relative density to the other. Nevertheless, differences between in vitro and in

vivo growth and metabolism owing to the nutritional composition of the growth environment may also help account for differences in the co-existence dynamics reported previously (*Filkins et al., 2015*; *Rajan and Saiman, 2002*).

It has been previously recognized that spatial organization facilitated by an undisturbed growth environment can facilitate the co-existence of bacteria, including *P. aeruginosa* and *S. aureus* (*Barraza and Whiteley, 2021*). Importantly, the production of HQNO from *P. aeruginosa* can impact the pattern of spatial organization of this co-cultured population (*Barraza and Whiteley, 2021*). While we did not explicitly quantify spatial organization experimentally owing to technical limitations of our microplate reader and microscope setups, in theory, co-culture in an undisturbed condition should facilitate the creation of spatial organization. We note that the use of microscopy would be a superior approach as compared to our ABM. We are currently working toward developing a system that would allow us to monitor the effect of disturbance on the spatial positioning of bacteria in real time. Nevertheless, we found that, in an undisturbed growth environment, the absolute growth ratio could predict the final population composition; increasing the absolute growth ratio increased the final density of *P. aeruginosa* relative to *S. aureus*. When disturbed periodically using a physical force, *S. aureus* dominated the final population composition. However, a clear trend between the absolute growth ratio and the final population composition was not observed when using a linear regression or WLS regression analysis. Our model suggests that these disturbances decrease the spatial overlap of both populations, and limit the amount of virulence factors that are sensed by *S. aureus*. Importantly, physical forces that are encountered where bacteria can infect hosts, including luminal areas (*Thomas et al., 2004*; *Lipowsky et al., 1978*), can fluctuate, which could serve to periodically alter the spatial positions, or overlap, between both populations. Thus, the degree to which a population of *S. aureus* and *P. aeruginosa* is disturbed may influence their ability to interact; this would impact the population composition. Importantly, we cannot rule out that increasing the amount of disturbances might have also impacted oxygenation of the medium, or caused unknown transcriptional responses in the bacteria. However, our model, and experiments using knockout strains, suggest that co-existence of *S. aureus* and *P. aeruginosa* in periodically disturbed environments is largely due to interactions governed by HQNO, and to a lesser extent, pyoverdine and pyochelin. While these disturbances have been previously shown to impact bacteria cooperation (*Wilson et al., 2017*) and the production of virulence factors (*Quinn et al., 2021*), to our knowledge, this is the study reporting that such disturbances can alter the population composition in a polymicrobial community. Thus, these disturbances represent a new method by which the composition, and potentially the functionality, of a polymicrobial community might be perturbed.

From an ecological perspective, our findings provide support for the intermediate disturbance hypothesis. This hypothesis suggests that the diversity of competing species is maximized at intermediate disturbance frequencies or intensities (*Connell, 1978*). While the intermediate disturbance hypothesis has faced criticism (*Fox, 2013*), evidence of its influence on biodiversity has been observed multiple times (*Roxburgh et al., 2004*), has helped to guide conservation programs (*Olff and Ritchie, 1998*), and can help explain the establishment of invasive species (*Catford et al., 2012*). Herein, our results provide evidence that further supports the intermediate disturbance hypothesis; disturbance at intermediate frequencies facilitates the co-existence of both *P. aeruginosa* and *S. aureus,* whereas in continually and undisturbed environments *P. aeruginosa* dominated the co-culture (*Figure 5*). This observation was found in both TSB medium and SCFM suggesting that it is robust in multiple environments. While previous work has also found that intermediate disturbance can influence the biodiversity of bacterial populations (*Brockhurst et al., 2007*), our work provides additional support for the intermediate disturbance hypothesis using two clinically important bacterial pathogens.

Co-culture of *S. aureus* and *P. aeruginosa* leads to changes in antibiotic susceptibility. For example, HQNO produced by *P. aeruginosa* interferes with ATP production in *S. aureus*. This causes a reduction in the metabolism of *S. aureus* and increases its resistance to aminoglycoside antibiotics. Importantly, a reduction in metabolism has been previously noted to decrease antibiotic efficacy across a wide range of bactericidal antibiotics (*Lopatkin et al., 2019*). Interestingly, HQNO can also increase membrane permeability of *S. aureus*, which increases the efficacy of antimicrobials such as chloroxylenol and fluoroquinolones (*Orazi et al., 2019*). Manipulation of absolute growth using different carbon sources may serve to rationally increase the efficacy of antibiotics. For example, carbon sources that cause *S. aureus* to have a significantly greater absolute growth relative to *P. aeruginosa* would limit the growth

of *P. aeruginosa*, and thus reduce the total concentration of HQNO produced. This would allow the metabolism of *S. aureus* to remain relatively high, thus promoting the efficacy of antibiotics, such as aminoglycosides. Conversely, promoting the production of *P. aeruginosa* by using carbon sources that increase its absolute growth relative to *S. aureus* would augment the production of HQNO. This could increase the membrane permeability of *S. aureus* found in biofilms, thus augmenting sensitivity to certain antibiotics, such as fluoroquinolones. Thus, rational manipulation of absolute growth in poly-microbial interactions may serve to differentially alter susceptibility to existing antibiotics.

# Methods

**Key resources table**

| Reagent type (species) or resource | Designation | Source or reference | Identifiers | Additional information |
|---|---|---|---|---|
| Strain (*S. aureus*) | RN4220 | BEI resources | N/A | Wildtype strain |
| Strain (*P. aeruginosa*) | PA14 | Lingchong You | N/A | Wildtype strain |
| Strain (*P. aeruginosa*) | PA14 (ΔpqsL) | Geisel School of Medicine at Dartmouth (Hanover, NH); *Orazi et al., 2019* | | Knockout strain |
| Strain (*P. aeruginosa*) | PA14 (ΔpvdA/ ΔpchE) | Geisel School of Medicine at Dartmouth (Hanover, NH); *Orazi et al., 2019* | | Knockout strain |
| Strain (*P. aeruginosa*) | PA14 (ΔpqsL /ΔpvdA/ ΔpchE) | Geisel School of Medicine at Dartmouth (Hanover, NH); *Orazi et al., 2019* | | Knockout strain |
| Commercial assay or kit (ATP assays) | BacTiter-Glo Microbial Cell Viability Assay | Promega | Cat. # G8230 | Used to measure ATP |
| Software, algorithm (MATLAB) | MATLAB with curve fitting and optimization toolboxes | MathWorks Inc. | N/A | Used to measure maximum growth rate |
| Software, algorithm (JMP Pro 16) | JMP Pro 16 | SAS Institute Inc. | N/A | Used for statistical analysis |
| Software, algorithm (Netlogo) | Netlogo | This paper; *Tisue and Wilensky, 2004* | | Used to measure spatial distribution |
| Other (Aeraseal film) | Aeraseal sealing membranes | Sigma Aldrich | Cat #A9224 | Used to cover 6 well plates |

## Strains and growth conditions

*P. aeruginosa* strain PA14 and *S. aureus* strain RN4220 were used in this study. *P. aeruginosa* mutants ΔpqsL, ΔpvdA/ΔpchE and ΔpqsL/ΔpvdA/ΔpchE were obtained from *Orazi et al., 2019*. Single colonies of *P. aeruginosa* and *S. aureus* isolated from Luria-Bertani (LB) agar medium (MP Biomedicals, Solon OH) were inoculated overnight in 3 mL of liquid LB medium and shaken (250 RPM and 37 °C) in 15 mL culture tubes (Genesee Scientific, Morrisville, NC). The following day, the cells were washed in fresh medium. For our assays, and as indicated in the text, we used tryptic soy broth (TSB) medium (Soytone (Thermo Fisher Scientific, Waltham, MA), Tryptone, Dipotassium Phosphate, and Sodium Chloride (VWR International, Radnor, PA)), modified AMM medium (*Rudin et al., 1974*; *Machado et al., 2019*) (162.6 mM NaCl (Thermo Fisher Scientific)), 40.2 mM KCl (Thermo Fisher Scientific), 10.8 mM $MgSO_4$ (BDH, VWR International), 30.3 mM (($NH_4)_2SO_4$) (Thermo Fisher Scientific), 2 mM $CaCl_2$ (Acros Organics, Geel, Belgium), 10.3 mM $KH_2PO_4$ (Thermo Fisher Scientific), 0.2 mM $FeSO_4•7H_2O$ (Alfa Aesar, Stoughton, MA), 0.4 mM $MnSO_4•4H_2O$ (Alfa Aesar), 0.3 mM citric acid (Thermo Fisher Scientific), 100 mM Tris (Thermo Fisher Scientific), buffered to pH 7.4 and SCFM (prepared as previously described *Palmer et al., 2007*). TSB medium was supplemented with 0.25% carbon source. AMM medium was supplemented with 0.04% carbon source and 0.01% casamino acids (Teknova, Hollister, CA). SCFM was supplemented with 0.054% carbon source, which matches the percentage of glucose used in this medium as previously described *Palmer et al., 2007*. Carbon sources tested in this manuscript were as follows: D(+) mannose (Acros Organics), glycerol (Acros Organics), D(+) ribose

(Acros Organics), 2- ketoglutaric acid (Alfa Aesar), D(+) galactose (Alfa Aesar), D-lactose (Alfa Aesar), sodium pyruvate (Alfa Aesar), sorbitol (Fisher Scientific), D-glucose (Fisher Scientific), D(+) lactic acid lithium salt (MP Biomedicals), succinic acid (Alfa Aesar), and sucrose (Sigma-Aldrich, St Louis, MO).

## Growth rate

Overnight cultures were washed once in growth medium lacking a carbon source and diluted 200-fold into 200 μL of fresh medium with a carbon source, both as indicated in the text. This was then placed in the wells of a 96-well plate. The medium was overlaid with 70 μL of mineral oil to prevent evaporation. Using a Perkin Elmer Victor X4 (Waltham, MA) plate reader, cell density was determined every 10 min (optical density at 600 nm [$OD_{600}$]) for approximately 15 hr at 37 °C. $OD_{600}$ values from cell-free medium were subtracted from all measurements prior to growth curve fitting. To help remove any artifacts of background and to ensure all data was initiated at the same starting point, we log-transformed and normalized to the initial minimum density. Together, this reduces the error when performing curve fitting. Growth rate was determined by fitting a logistic curve to the data using a custom MATLAB (MathWorks Inc, Natick, MA) code (*Diaz-Tang et al., 2022*). Average residuals are shown in *Supplementary file 8*; lower residual values indicate a stronger fit between our experimental data and the curve fitting. We used an average residual of less than one to indicate an acceptable fit.

## Determining the Concentration of ATP

Overnight cultures were washed once with fresh medium lacking a carbon source. Bacteria were then diluted 10-fold in fresh medium supplemented with a carbon source in a 6-well cell culture plate (Genesee Scientific). The plate was overlaid with two AeraSeal sealing membranes (Sigma-Aldrich). For TSB medium, the plates were shaken at 110 RPM pre-set to 37 °C for 3 hr whereupon the bacteria entered log phase growth; gentle shaking allowed ATP measurement to fit a wider range of conditions where the culture was disturbed (*Figure 5*). For AMM medium and SCFM, the plates were incubated in a stationary incubator at 37 °C for 3 hr; the stationary condition better matched the majority of the conditions under which these experiments were performed. A total of 100 μL of these monocultures was added to the wells of an opaqued walled 96-well plate and ATP was measured using a biolumines-cent kit as recommended by the manufacturer (BacTiter-Glo Microbial Cell Viability Assay, Promega, Madison, WI). An ATP standard curve prepared using purified ATP (Sigma Aldrich) was used to determine the concentration of ATP (*Diaz-Tang et al., 2022*). All ATP measurements were normalized to cell density ($OD_{600}$) and converted to concentration using a standard curve prepared using purified ATP.

## Co-culture assays

Single colonies of *P. aeruginosa* and *S. aureus* were grown overnight and washed with fresh medium. Bacteria were then diluted to an $OD_{600}$ of 0.1, which leads to an equal initial density between both strains (*Figure 2—figure supplement 1*). The normalized cells were then diluted 100-fold in 6-well cell culture plates, overlaid with two AeraSeal sealing membranes, and placed in an undisturbed incubator for 30 min at 37 °C. For most assays, the monocultures of *P. aeruginosa* and *S. aureus* were then combined in a 1:1 ratio to a volume of 3 mL, and re-sealed with AeraSeal film. We altered the initial bacterial ratio by scaling the addition of *P. aeruginosa* monoculture to the co-culture to accommodate for a total volume of 3 mL. For the stationary (undisturbed) and continuously disturbed conditions, the plates were either placed in a stationary incubator at 37 °C or shaken at 110 RPM at 37 °C, respectively, for 24 hr. The plates that were periodically shaken were placed in a Victor X4 plate reader pre-set to 37 °C (slow setting, frequency, 10 sr per shaking event, orbital shaking feature, radius = 5 mm) at the indicated frequency. For experiments performed in an anaerobic environment, we overlaid the cultures in the cultures in the 96-well plate with 3 mL of mineral oil and we did not add AeraSeal membranes. After 24 hr of growth, 1 mL of culture was removed and washed with 1 x phosphate buffered saline (PBS) (Thermo Fisher Scientific). The culture was then sonicated in a water bath for 2 min to reduce clumping of *S. aureus*. We then performed a serial dilution and selective plating on mannitol salt agar (Becton, Dickinson and Company, Franklin Lakes, NJ, selects for *S. aureus*) and cetrimide (Oxoid, Basingstoke, England, selects for *P. aeruginosa*). For experiments performed in TSB or AMM medium we determined the number of colony forming units (CFU) grown on the selective agar medium following ~20 hr of incubation at 37 °C. For experiments performed in SCFM, colonies

were counted after 24–48 hr of incubation at 37 °C. In this medium, cultures grown in some carbon sources (ribose, pyruvate, lactic acid) required additional incubation time so that colonies would grow on the MSA plates.

## Statistical analysis

Statistical analysis as indicated in the text or figure legend. Unpaired t-tests (unequal variance) were performed using Microsoft Excel (Redmond, WA). Additional tests were performed in JMP Pro 16 (SAS Institute Inc, Cary, NC). We took the natural logarithm of the ratio of *P. aeruginosa* to *S. aureus* to linearize the data; these values were used for linear regressions. To test if the ratios themselves differed within a dataset using a Mann-Whitney or Kruskal-Wallis, we used the raw ratio data that was not ln transformed. When performing a weighted least squares (WLS) regression (***Strutz, 2011***), we first converted standard deviation to variance, and then used inverse variance to weight each data point. Thus, data points with larger standard deviations have less weight when performing a linear regression. A Shapiro-Wilk test was used to assess normality. P values from all statistical tests (including tests that were performed to determine data distribution) were corrected for Type I errors using a Benjamini-Hochberg Procedure (***Benjamini and Hochberg, 1995***) with an 8.5% discovery rate. With this correction, p values that were <0.0472 were considered significant. Standard error of the mean (SEM) for absolute growth and the absolute growth ratio was determined using a propagation of errors approach.

## Mathematical modeling - ODE

We modeled the interaction between *S. aureus* and *P. aeruginosa* using three ordinary differential equations that capture the synthesis of virulence factors that affect metabolism in *S. aureus* (***Equation 1***), growth of *P. aeruginosa* (***Equation 2***), and growth of *S. aureus* (***Equation 3***).

$$\frac{d[vir]}{dt} = \frac{k_p C_p}{K + C_p} - d_v [vir] \tag{1}$$

$$\frac{dC_p}{dt} = \mu_p \varepsilon_p C_p \left(1 - \frac{C_p}{C_m}\right) - d_c C_p \tag{2}$$

$$\frac{dC_s}{dt} = \mu_s \frac{\varepsilon_s}{1 + \delta[vir]} C_s \left(1 - \frac{C_s}{C_m}\right) - d_c C_s \tag{3}$$

where $k_p$ represents the maximum synthesis rate of virulence factors that affect metabolism in *S. aureus* (*vir*), $d_v$ represents the degradation rate of virulence factors, $C_p$ and $C_s$ represent the density of *P. aeruginosa* and *S. aureus*, respectively, $\mu_p$ and $\mu_s$ represents the maximum growth rate of *P. aeruginosa* and *S. aureus*, respectively, $\varepsilon_p$ and $\varepsilon_s$ represents metabolism of *P. aeruginosa* and *S. aureus*, respectively $C_m$ represents the carrying capacity of the medium normalized to 1, $d_c$ represents death rate of bacteria, and $\delta$ represents the amount of spatial overlap in the system. The synthesis of *vir* is modeled using a modified Hill Equation and is dependent upon cell density as previous work has indicated non-linear and density-dependent production of HQNO and pyoverdine (***Déziel et al., 2004***; ***Julou et al., 2013***). The growth term of ***Equations 2; 3*** is modeled using a simplified version of Monod's growth. Maximum growth rate, $\mu$, is scaled by the metabolic state, $\varepsilon$, of the cell; the product of these two variables determines the effective growth, which is used to inform the logistic growth term. $\varepsilon$ approximates a maintenance coefficient. The growth term of *S. aureus* (***Equation 3***) is scaled by the concentration of virulence factors that affect metabolism. As observed in previous work, increasing concentrations of HQNO, pyoverdine and pyochelin decrease metabolism, which in turn decreases growth rate. An increase in the concentration of these virulence factors would reduce the value of $\varepsilon$. This in turn would reduce the entire growth term ($\mu$ x $\varepsilon$), thus capturing the dynamics of previous experimental work. $\delta$ scales the concentration of virulence factors that are sensed by *S. aureus*; it is a simplification of the spatial distribution of *S. aureus* and *P. aeruginosa* during disturbance. Lower values of $\delta$ indicate reduced spatial overlap between *S. aureus* and *P. aeruginosa*.

## Parameter estimation -ODE

For simulations where growth rate ($\mu$) was not varied, we estimated the growth rate of both bacterial species using the average growth rate found in the study (***Figure 1B***, ***Figure 3A***, ***Figure 3F*** and *S. aureus* = 0.40 +/- 0.19; *P. aeruginosa* = 0.3 +/- 0.045). We also ensured that the growth rate of *S. aureus* was greater than that of *P. aeruginosa*, which was consistent with the general trends in our

experiments (*Figure 1B*). The range of values of $\varepsilon$ were estimated using previously published maintenance coefficients (*Wallace and Holms, 1986*; *Sauer et al., 1996*; *Stouthamer and Bettenhaussen, 1975*). The value of $k_p$ was estimated as it represents a lumped term that captures the synthesis of three different virulence factors; pyoverdine, HQNO, and pyochelin. The average amount of pyoverdine synthesized by *P. aeruginosa* has been reported to be approximately 0.3 µM/hr (*Julou et al., 2013*). The approximate amount of HQNO synthesized by a population of *P. aeruginosa* is approximately 3500 µM/hr (*Déziel et al., 2004*). Finally, the amount of pyochelin synthesized is approximately 0.002–0.02 µM/hr (*Serino et al., 1997*). Given the wide range of synthesis values, and the simplification of our $k_p$ term, we approximated $k_p$ of 1 µM/hr. Increasing or decreasing this value tenfold maintains the qualitative trends in our simulations (*Figure 2—figure supplement 2*). To estimate the value of $d_v$, we considered previously reported degradation rates of pyoverdine (dilution rate of approximately 0.5 /hr *Jin et al., 2018*) and HQNO (decay of approximately 2 mg/mL over 10 hr, or ~0.01 / hr *Lépine et al., 2003*). As above, given the wide range of dilution/decay rates, we estimated a central value of 0.1 /hr; increasing or decreasing this value 10-fold does not change the qualitative nature of our predictions (*Figure 2—figure supplement 2*). We estimated the order of magnitude of the cell death ($d_c$) using previously published data (*Wang et al., 2010*). We fit the value of $K$ to our experimental data. Values of $\delta$ were benchmarked to a value of 1, which indicates near complete spatial overlap between *P. aeruginosa* and *S. aureus*. Reducing the value of $\delta$ served to decrease the spatial overlap of both species, thus reducing the effect of these virulence factors. Trends in $\delta$ were approximated using agent-based modeling (see below, *Figure 4A*). Simulations were performed for $t$=24 hr, which is the same amount of time over which our experiments occurred. The general trend of our simulations remains the same when parameters are increased or decreased 10-fold from the base set of parameter values (*Supplementary file 2*).

## Mathematical modeling – agent-based model

We constructed an agent-based model (ABM) in the program Netlogo (*Tisue and Wilensky, 2004*) to better understand the amount of spatial overlap of two bacterial populations under various disturbance conditions. To coincide with our experimental approach, the model begins with equal-sized populations of *S. aureus* and *P. aeruginosa* cells (agents) placed in completely overlapping distributions at the center of the virtual world and spread-out based on a normal distribution. The virtual world is a two-dimensional grid comprised of 10,201 patches (101 patches vertically x 101 patches horizontally). In each round, individual agents reproduce to gradually grow the population logistically ($r$=0.5) from its initial size to reach the final carry capacity of 200,000 agents. While there are no antagonistic interactions between the populations, growth is still limited by local density as each patch has a carrying capacity (of overall carrying capacity divided by number of patches). Therefore, reproduction is diminished in crowded patches and more favorable in less occupied patches. To mimic periodic spatial disturbances, each time step all cells are displaced (called 'distance travelled') in random directions at a percentage distance of the virtual world (base case is 1% or one patch). These disturbances were varied from occurring once each round, to up to 50-times each round, or not at all. Simulations were iterated under given disturbance conditions for 24-rounds, to resemble 24 hr laboratory experiments. The purpose of these simulations was to observe the extent that disturbance alone influences the extent of overlap of competing populations. The amount of overlap was represented by the variable, cohabitation, which is the observed frequency of patches hosting agents from both populations minus expected random frequency of co-habitation. Specifically, this was calculated as the number of observed co-occupied/total occupied patches, subtracted by the product of frequency of patches occupied by *S. aureus* agents and frequency of patches occupied by *P. aeruginosa*. Therefore, a positive value indicates that the two populations cohabit patches at a frequency greater than expected randomly, and a negative value indicates that individuals are less likely to co-occupy patches than expected randomly. For each simulation, this value was averaged across all 24-rounds.

## Acknowledgements

This research was sponsored by the Army Research Office and was accomplished under Grant Number W911NF-18-1-0443. The views and conclusions contained in this document are those of the authors and should not be interpreted as representing the official policies, either expressed or implied, of the Army Research Office or the U.S. Government. The U.S. Government is authorized to reproduce

and distribute reprints for Government purposes notwithstanding any copyright notation herein. *S. aureus* strain RN4220 was obtained through BEI Resources, NIAID, NIH. The Δ*pqsL*, Δ*pvdA*/Δ*pchE* and Δ*pqsL*/Δ*pvdA*/Δ*pchE* knockout strains were a kind gift from Dr. George A O'Toole.

## Additional information

### Funding

| Funder | Grant reference number | Author |
|---|---|---|
| Army Research Office | W911NF-18-1-0443 | Camryn Pajon<br>Marla C Fortoul<br>Gabriela Diaz-Tang<br>Estefania Marin Meneses<br>Ariane R Kalifa<br>Elinor Sevy<br>Taniya Mariah<br>Brandon Toscan<br>Maili Marcelin<br>Daniella M Hernandez<br>Melissa M Marzouk<br>Omar Tonsi Eldakar<br>Robert P Smith |

The funders had no role in study design, data collection and interpretation, or the decision to submit the work for publication.

### Author contributions

Camryn Pajon, Marla C Fortoul, Data curation, Formal analysis, Investigation, Methodology, Writing – original draft, Writing – review and editing; Gabriela Diaz-Tang, Estefania Marin Meneses, Formal analysis, Investigation, Writing – review and editing; Ariane R Kalifa, Elinor Sevy, Maili Marcelin, Daniella M Hernandez, Melissa M Marzouk, Investigation, Writing – review and editing; Taniya Mariah, Formal analysis, Methodology; Brandon Toscan, Investigation; Allison J Lopatkin, Software, Methodology, Writing – review and editing; Omar Tonsi Eldakar, Conceptualization, Software, Methodology, Writing – original draft, Writing – review and editing; Robert P Smith, Conceptualization, Data curation, Formal analysis, Supervision, Funding acquisition, Investigation, Methodology, Writing – original draft, Project administration, Writing – review and editing

### Author ORCIDs

Robert P Smith ⬡ http://orcid.org/0000-0003-2744-7390

### Decision letter and Author response

Decision letter https://doi.org/10.7554/eLife.83664.sa1
Author response https://doi.org/10.7554/eLife.83664.sa2

## Additional files

### Supplementary files

• Supplementary file 1. Statisical analysis for *Figure 1*. (a) P values for data presented in *Figure 1*. *n* represents the number of biological replicates. To calculate absolute growth, all biological replicates from maximum growth rate and [ATP] were used. Thus, the value of *n* for absolute growth represents the smallest number of biological replicates included in the calculation.

• Supplementary file 2. Parameter set and statistical analysis for *Figure 2*. (a) Base parameter set used in our mathematical model (*Equations 1–3*). (b) P values for data presented in *Figure 2*. *n* represents the number of biological replicates. Shapiro-Wilk for data sets in *Figure 2A and B*, $P<0.0001$. In *Figure 2B and a* Dunn test for joint rank was used to compare final density ratios. The final density ratio of the wildtype *P. aeruginosa* strain co-cultured with *S. aureus* was used as the control. TSB supplemented with glucose was used for these experiments.

• Supplementary file 3. Statisical analysis for *Figure 2—figure supplement 1*. (a) P values for data presented in *Figure 2—figure supplement 1*, panel H. Shapiro-Wilk for final density ratios,

$P$=0.0004; for final bacterial densities, $P$<0.0001. $n$ represents the number of biological replicates.

• Supplementary file 4. Statistical analysis for *Figure 3*. (a) P values for data presented in *Figure 3*. $n$ represents the number of biological replicates. To calculate absolute growth, all biological replicates from maximum growth rate and ATP were used. Thus, the value of $n$ for absolute growth represents the smallest number of biological replicates included in the calculation. Two-tailed t-tests were used for [ATP], growth rate and absolute growth measurements. A Mann-Whitney (Shapiro-Wilk, $P$<0.0001) was used to assess differences in the final density of *P. aeruginosa* and *S. aureus* (*Figure 3E and J*).

• Supplementary file 5. P values for data presented in *Figure 4*. Shapiro-Wilk, $P$≤0.0001 for all carbon sources. $n$ represents the number of biological replicates.

• Supplementary file 6. Statistical analysis for *Figure 5*. (a) P values for data presented in *Figure 5E, G and F*. Shapiro-Wilk for all data sets in this figure; $P$≤0.019. $n$ represents the number of biological replicates. (b) P value for data presented in *Figure 5F*, right panel. Shapiro-Wilk for final density ratios, $P$<0.0001. Shapiro-Wilk for final bacterial densities, $P$=0.0190. $n$ represents the number of biological replicates.

• Supplementary file 7. Statistical analysis for *Figure 5—figure supplement 3*. (a) P values for data presented in *Figure 5—figure supplement 3*. Shapiro-Wilk for final density ratio and for bacterial densities, $P$<0.0001.

• Supplementary file 8. Residual values for growth curve fitting. (a) Average residual values for all curve fitting and biological replicates. Curve fitting was used to determine maximum growth rate in *Figure 1B* (TSB medium). (b) Average residual values for all curve fitting and biological replicates. Curve fitting was used to determine maximum growth rate in *Figure 3A* (AMM medium). (c) Average residual values for all curve fitting and biological replicates. Curve fitting was used to determine maximum growth rate in *Figure 3F* (SCFM medium).

• MDAR checklist

## Data availability

All raw experimental data is currently deposited in Dryad at https://doi.org/10.5061/dryad.fn2z34tz3.

The following dataset was generated:

| Author(s) | Year | Dataset title | Dataset URL | Database and Identifier |
|---|---|---|---|---|
| Smith RP | 2023 | Interactions between metabolism and growth can determine the co-existence of *Staphylococcus aureus* and *Pseudomonas aeruginosa* | https://doi.org/10.5061/dryad.fn2z34tz3 | Dryad Digital Repository, 10.5061/dryad.fn2z34tz3 |

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
