## [Editor Report]

How the pathogens *Pseudomonas* aeruginosa and *Staphylococcus aureus* compete and co-occur within opportunistic infections is an important problem, but the major drivers of these interactions remain unclear. Here the authors contribute a fundamental advance by defining parameters that predict the coexistence of these microbes using their absolute growth in various nutritional conditions, which could explain the dominance of one or the other during infections. Within a defined context, this valuable study provides solid support for a novel framework in which to evaluate this clinically important species interaction.

---

## [Decision Letter]

**Decision letter after peer review:**

Thank you for submitting your article "Interactions between metabolism and growth can determine the co-existence of *Staphylococcus aureus* and *Pseudomonas aeruginosa*" for consideration by *eLife*. Your article has been reviewed by 3 peer reviewers, one of whom is a member of our Board of Reviewing Editors, and the evaluation has been overseen by Aleksandra Walczak as the Senior Editor. The reviewers have opted to remain anonymous.

Essential revisions:

1) The correlation between ATP levels and relative growth rates for Staph is interesting and informative, but the relatively flat growth responses and ATP concentration responses of Pa are because they are primarily eating the amino acids in the TSB. This becomes a major limiter for understanding carbon source impacts on Pa, while still potentially valid for Staph in an environment where Pa is mostly growing on amino acids.

The predictive premise of the model derived from the TSB-based experiments should be tested in two different (non-TSB, non-LB, non-BHI) media formulations and the relative relationship between absolute growth observed. Currently, the study only retests competitive outcomes in the same media from which the prediction was derived.

2) Virulence mutants of Pa should be grown in each media condition to determine if there is a growth deficit of the mutant compared to wt. If this is already in the text it was not clear and should be highlighted in the text as a control.

3) More explicit mention of the intermediate disturbance hypothesis and possible underlying effects of disturbance would be worthwhile, to connect readers to this foundational concept in ecology. In addition, a large literature exploring interactions between these species is cited but under-discussed.

*Reviewer #1 (Recommendations for the authors):*

The analysis of virulence factors is somewhat less thorough given that their induction is often tied to many factors beyond nutrition, but it is remarkable how well fueling explains who wins and loses. Additional discussion of the wide range of inducers of virulence factors is worthwhile.

More explicit mention of the intermediate disturbance hypothesis and possible underlying effects of disturbance are worthwhile, to connect readers to this foundational concept in ecology.

*Reviewer #2 (Recommendations for the authors):*

I have some recommendations for additional required data/controls and comments:

- line 282 – authors discuss 'tipping point' in their model, without the mention of quorum sensing which is likely what this is modeling. E.g a threshold cell density. This should be discussed in the context of the relevant data here and in the discussion. many of the virulence factors assessed here are also QS dependent. A way to test this experimentally may be to use a dual carbon source of casamino acids and those selected previously to control an initial density or tipping point (for future studies, Darch 2012 PNAS).

- Virulence mutants of Pa should be grown in each media condition to determine if there is a growth deficit to mutation compared to wt. If this is already in the text it was not clear and should be highlighted in the text as a control.

- Each of the virulence mutants should be complemented and the experiments performed again to show the phenotype is lost in co-culture with SA.

- Overall the paper is very well written and the figures are presented nicely. There are some spots where italics are needed for organism names etc.

*Reviewer #3 (Recommendations for the authors):*

I have no additional major comments other than those described in the Public Review. If this study had been conducted within the SCFM2 media background via constituent substitution it could have been groundbreaking and immediately relevant to the field. Occurring, as it does, in TSB, it is subject to too many caveats to enhance or supplant existing work demonstrating situational contributions to co-culture.

---

## [Author Response]

Essential revisions:1) The correlation between ATP levels and relative growth rates for Staph is interesting and informative, but the relatively flat growth responses and ATP concentration responses of Pa are because they are primarily eating the amino acids in the TSB. This becomes a major limiter for understanding carbon source impacts on Pa, while still potentially valid for Staph in an environment where Pa is mostly growing on amino acids.The predictive premise of the model derived from the TSB-based experiments should be tested in two different (non-TSB, non-LB, non-BHI) media formulations and the relative relationship between absolute growth observed. Currently, the study only retests competitive outcomes in the same media from which the prediction was derived.

We have tested the relationship between the final density ratio of *P. aeruginosa* to *S. aureus* and the ratio of absolute growth in synthetic minimal medium (AMM), which allows for the growth of *S. aureus* (see more below in response to Reviewer 3) and synthesis cystic fibrosis medium (SCFM). In both cases, we continue to find strong and significant linear relationships between the final population composition and the ratio of absolute growth. This is consistent when we applied either a simple linear regression or a weighted least squares (WLS) regression, which considers error on the y-axis (in this case error in the final density ratio). This new data is presented in Figure 3.

In addition, we also confirmed that periodically disturbing co-cultured populations grown in SCFM allows *S. aureus* to gain a competitive advantage at intermediately, but not infrequent or frequently, disturbed environments. This data is noted in the Results section and is presented in Figure 5 —figure supplement 3.

2) Virulence mutants of Pa should be grown in each media condition to determine if there is a growth deficit of the mutant compared to wt. If this is already in the text it was not clear and should be highlighted in the text as a control.

We measured the growth rate of all three mutants and compared it to the wildtype growth rate in all growth conditions presented in our original manuscript. Using a Dunnett’s test with the wildtype growth rate as the control, we did not a find a significant decrease in the growth rate of the mutant strains across all conditions tested. Thus, a reduction in growth rate cannot account for the differences in final density ratio when using the knockout strains. This data is highlighted in the Results section and shown in Figure 2 —figure supplement 1, panels K-L. All P values and biological replicate numbers are found in Supplementary File 2.

3) More explicit mention of the intermediate disturbance hypothesis and possible underlying effects of disturbance would be worthwhile, to connect readers to this foundational concept in ecology. In addition, a large literature exploring interactions between these species is cited but under-discussed.

1)We have made mention of the intermediate disturbance hypothesis in the revised abstract. We have also discussed the implications of our finding for this hypothesis in the discussion where we write:

“From an ecological perspective, our findings provide support for the intermediate disturbance hypothesis. This hypothesis suggests that the diversity of competing species is maximized at intermediate disturbance frequencies or intensities [1]. While the intermediate disturbance hypothesis has faced criticism [2], evidence of its influence on biodiversity has been observed multiple times [3], has helped to guide conservation programs [4], and can help explain the establishment of invasive species [5]. Herein, our results provide evidence that further supports the intermediate disturbance hypothesis; disturbance at intermediate frequencies facilitates the co-existence of both *P. aeruginosa* and *S. aureus* whereas in continually and undisturbed environments *P. aeruginosa* dominated the co-culture (Figure 5). This observation was found in both TSB medium and SCFM suggesting that it is robust in multiple environments. While previous work has also found that intermediate disturbance can influence the biodiversity of bacterial populations [6], our work provides additional support for the intermediate disturbance hypothesis using two clinically important bacterial pathogens.”

2) We have expanded the information provided on the interactions between *P. aeruginosa* and *S. aureus.* In the introduction, we have expanded on the mechanisms by which both species interact and its impact on growth and metabolism. This section now reads:

“…Their co-culture induces interspecies competition, which enhances antibiotic resistance and virulence. In the presence of *P. aeruginosa*, *S. aureus* upregulates the expression of virulence factors including staphylococcal protein A, α-hemolysis and Panton-Valentine leucocidin [7]. While *S. aureus* can initially aid in the establishment of the *P. aeruginosa* population [8], production of N-acetylglucosamine from *S. aureus* augments the virulence of *P. aeruginosa* by upregulating expression of multiple secreted virulence factors. Many of these are regulated by quorum sensing and include pyocyanin [9], pyoverdine [10], 2-Heptyl-4-hydroxyquinoline N-oxide [11], and 4-hydroxy-2-heptylquinoline N-oxide (HQNO)[12-15]. Some of these virulence factors reduce metabolism in *S. aureus* through multiple mechanisms. Pyocyanin induces the production of reactive oxygen species in *S. aureus* by reacting with menadione, which results in oxidative damage [16]. Pyocyanin has been found to interfere with electron transport, reducing respiration [17]. The siderophores pyoverdine and pyochelin drive *S. aureus* into fermentation [18]. While exposure to HQNO can initially promote biofilm formation in *S. aureus* [19], it eventually interferes with the activity of cytochrome b, which reduces ATP production [20]. N-3-oxo-dodecanoyl-L-homoserine lactone (3OC12HSL) produced by *P. aeruginosa* for the purposes of quorum sensing has been shown to reduce growth of *S. aureus* [21]. Overall, multiple virulence factors interfere with growth and metabolism in *S. aureus*, ultimately leading to facultative respiration and increased antibiotic tolerance [22]...”

In the Discussion section, and in response to a comment by reviewer 3 (below), we have also expanded on the interactions that facilitate co-existence of both species. Specifically, we write:

“Given the clinical importance of co-infection with both *P. aeruginosa* and *S. aureus*, multiple previous studies have identified mechanisms of co-existence. Indeed, long term co-existence of both species can result in physiological changes that reduce their competitive interactions. Strains of *P. aeruginosa* isolated from patients that enter into a mucoid state show reduced production of siderophores, pyocyanin, rhamnolipids and HQNO, which facilitates the survival of *S. aureus* [23, 24]. These strains can also overproduce the polysaccharide alginate, which in itself is sufficient to decrease the production of these virulence factors. Moreover, exogenously supplied alginate can reduce the production of pyoverdine and expression from the PQS quorum sensing system, which is responsible for the production of HQNO [25]. Changes in the physiology of *S. aureus* can also facilitate co-existence. Strains of *S. aureus* isolated from patients with cystic fibrosis show multiple changes in the abundance of proteins including super oxide dismutase, the GroEL chaperone protein, and multiple surface associated proteins [26]. Interestingly, the majority of proteins that show changes in abundance in *S. aureus* are related to central metabolism, which is consistent with our findings demonstrating that metabolism can influence the co-existence of both species. While it is unclear as to how long-term co-culture would affect the ratio of absolute growth, our findings provide an additional mechanism that can determine the co-existence of these bacterial species.”

Overall, we hope that all three Reviewers agree that we have satisfactorily addressed the essential revisions.

Reviewer #1 (Recommendations for the authors):The analysis of virulence factors is somewhat less thorough given that their induction is often tied to many factors beyond nutrition, but it is remarkable how well fueling explains who wins and loses. Additional discussion of the wide range of inducers of virulence factors is worthwhile.

To address this comment, we have included information of how quorum sensing also leads to the induction of virulence factor expression (which also helped to address a comment made by Reviewer 2). We have made this modification in both the introduction and the Discussion sections.

Introduction: “While *S. aureus* can initially aid in the establishment of the *P. aeruginosa* population [8], production of N-acetylglucosamine from *S. aureus* augments the virulence of *P. aeruginosa* by upregulating expression of multiple secreted virulence factors. Many of these are regulated by quorum sensing and include pyocyanin [9], pyoverdine [10], 2-Heptyl-4-hydroxyquinoline N-oxide [11], and 4-hydroxy-2-heptylquinoline N-oxide (HQNO)[12-15].

Discussion: “The absolute growth ratio also determines the tipping point of the co-cultured population. Importantly, many virulence factors produced by *P. aeruginosa* (e.g., HQNO, pyocyanin) that impact growth and metabolism in *S. aureus* are regulated by quorum sensing. Quorum sensing involves the production and exchange of small diffusible molecules, called autoinducers [27]. Since each *P. aeruginosa* bacterium produces autoinducers, the concentration of autoinducer increases as a function of cell density. At a sufficiently high concentration, the autoinducer will activate expression of multiple virulence factors in *P. aeruginosa*, which begin to impact growth and metabolism in *S. aureus*. Given this density-dependent activation of virulence factors, it is likely that the tipping point in our systems is being determined by quorum sensing, although we did not explicitly investigate this hypothesis in this manuscript.”

More explicit mention of the intermediate disturbance hypothesis and possible underlying effects of disturbance are worthwhile, to connect readers to this foundational concept in ecology.

As noted in our response in the ‘essential revisions’ section, we have included mention of the intermediate disturbance hypothesis in the abstract and have discussed our findings in the scope of this hypothesis in the Discussion section.

Reviewer #2 (Recommendations for the authors):I have some recommendations for additional required data/controls and comments:- line 282 – authors discuss 'tipping point' in their model, without the mention of quorum sensing which is likely what this is modeling. E.g a threshold cell density. This should be discussed in the context of the relevant data here and in the discussion. many of the virulence factors assessed here are also QS dependent. A way to test this experimentally may be to use a dual carbon source of casamino acids and those selected previously to control an initial density or tipping point (for future studies, Darch 2012 PNAS).

We would like to thank the reviewer for the suggestion for future work.

We have also modified the Discussion section to acknowledge that quorum sensing is likely playing a role in the tipping point analysis as follows:

“The absolute growth ratio also determines the tipping point of the co-cultured population. Importantly, many virulence factors produced by *P. aeruginosa* (e.g., HQNO, pyocyanin) that impact growth and metabolism in *S. aureus* are regulated by quorum sensing. Quorum sensing involves the production and exchange of small diffusible molecules, called autoinducers [27]. Since each *P. aeruginosa* bacterium produces autoinducers, the concentration of autoinducer increases as a function of cell density. At a sufficiently high concentration, the autoinducer will activate expression of multiple virulence factors in *P. aeruginosa*, which begin to impact growth and metabolism in *S. aureus*. Given this density-dependent activation of virulence factors, it is likely that the tipping point in our systems is being determined by quorum sensing, although we did not explicitly investigate this hypothesis in this manuscript. ”

- Virulence mutants of Pa should be grown in each media condition to determine if there is a growth deficit to mutation compared to wt. If this is already in the text it was not clear and should be highlighted in the text as a control.

This was something that we initially overlooked in our original manuscript. However, as noted in response to the ‘essential revisions’, we have indeed performed this analysis. We found that there was not a significant difference in growth rates between all three knockout strains and the wildtype strain. This is noted in the Results section and this new data is presented in Figure 2 —figure supplement 2.

- Each of the virulence mutants should be complemented and the experiments performed again to show the phenotype is lost in co-culture with SA.

We appreciate this suggestion by the reviewer. We chose not to perform these experiments as there was no difference in growth rate between the knockout strains and the wildtype strains. Moreover, multiple previous studies have demonstrated that the removal of these virulence factors from *P. aeruginosa* allows *S. aureus* to grow to higher cell densities when in co-culture. Specifically, O’Toole et al. (2015) demonstrated that *S. aureus* killing is significantly reduced in co-culture with *P. aeruginosa* Δ*pqsL* compared to wildtype *P. aeruginosa* PA14[18]. This was also consistent when *S. aureus* was co-cultured with *P. aeruginosa* knockout strains *ΔpvdA/ΔpchE* and Δ*pqsL* /Δ*pvdA/*Δ*pchE.* These findings were also echoed by Barraza and Whiteley (2021) who demonstrated that *P. aeruginosa* strain *ΔpqsL* mutant lysed *S. aureus* to a lesser degree than wildtype *P. aeruginosa* under well-mixed conditions in co-culture [28]. Overall, these studies support the conclusion that *P. aeruginosa* is dependent upon the HQNO pathway and its major siderophores to suppress *S. aureus* growth in co-culture.

- Overall the paper is very well written and the figures are presented nicely. There are some spots where italics are needed for organism names etc.

Thank you for your kind words and we apologize for this oversight. We have triple checked that all species names in the text are italicized. However, abbreviated genus names that serve as general identifiers in the figures remain unitalicized.

Reviewer #3 (Recommendations for the authors):I have no additional major comments other than those described in the Public Review. If this study had been conducted within the SCFM2 media background via constituent substitution it could have been groundbreaking and immediately relevant to the field. Occurring, as it does, in TSB, it is subject to too many caveats to enhance or supplant existing work demonstrating situational contributions to co-culture.

We hope that our use of SCFM, along with the additional modifications to our manuscript described here, has increased the relevance of our work in the field.

References

1. Connell JH. Diversity in tropical rain forests and coral reefs: high diversity of trees and corals is maintained only in a nonequilibrium state. Science. 1978;199(4335):1302-10.

2. Fox JW. The intermediate disturbance hypothesis should be abandoned. Trends in ecology and evolution. 2013;28(2):86-92.

3. Roxburgh SH, Shea K, Wilson JB. The intermediate disturbance hypothesis: patch dynamics and mechanisms of species coexistence. Ecology. 2004;85(2):359-71.

4. Olff H, Ritchie ME. Effects of herbivores on grassland plant diversity. Trends in ecology and evolution. 1998;13(7):261-5.

5. Catford JA, Daehler CC, Murphy HT, Sheppard AW, Hardesty BD, Westcott DA, et al. The intermediate disturbance hypothesis and plant invasions: Implications for species richness and management. Perspectives in plant ecology, evolution and systematics. 2012;14(3):231-41.

6. Brockhurst MA, Buckling A, Gardner A. Cooperation peaks at intermediate disturbance. Current Biology. 2007;17(9):761-5.

7. Pastar I, Nusbaum AG, Gil J, Patel SB, Chen J, Valdes J, et al. Interactions of methicillin resistant *Staphylococcus aureus* USA300 and *Pseudomonas* aeruginosa in polymicrobial wound infection. PloS one. 2013;8(2):e56846.

8. Alves PM, Al-Badi E, Withycombe C, Jones PM, Purdy KJ, Maddocks SE. Interaction between *Staphylococcus aureus* and *Pseudomonas* aeruginosa is beneficial for colonisation and pathogenicity in a mixed biofilm. Pathogens and disease. 2018;76(1):fty003.

9. Korgaonkar AK, Whiteley M. *Pseudomonas aeruginosa* enhances production of an antimicrobial in response to N-acetylglucosamine and peptidoglycan. Journal of bacteriology. 2011;193(4):909-17.

10. Orazi G, Ruoff KL, O’Toole GA. *Pseudomonas* aeruginosa increases the sensitivity of biofilm-grown *Staphylococcus aureus* to membrane-targeting antiseptics and antibiotics. MBio. 2019;10(4):e01501-19.

11. Machan ZA, Taylor GW, Pitt TL, Cole PJ, Wilson R. 2-Heptyl-4-hydroxyquinoline N-oxide, an antistaphylococcal agent produced by *Pseudomonas aeruginosa*. Journal of Antimicrobial Chemotherapy. 1992;30(5):615-23.

12. Li H, Li X, Wang Z, Fu Y, Ai Q, Dong Y, et al. Autoinducer-2 regulates *Pseudomonas aeruginosa* PAO1 biofilm formation and virulence production in a dose-dependent manner. BMC microbiology. 2015;15(1):1-8.

13. Déziel E, Lépine F, Milot S, He J, Mindrinos MN, Tompkins RG, et al. Analysis of *Pseudomonas aeruginosa* 4-hydroxy-2-alkylquinolines (HAQs) reveals a role for 4-hydroxy-2-heptylquinoline in cell-to-cell communication. Proceedings of the National Academy of Sciences. 2004;101(5):1339-44.

14. Nadal Jimenez P, Koch G, Thompson JA, Xavier KB, Cool RH, Quax WJ. The multiple signaling systems regulating virulence in *Pseudomonas aeruginosa*. Microbiology and Molecular Biology Reviews. 2012;76(1):46-65.

15. Korgaonkar A, Trivedi U, Rumbaugh KP, Whiteley M. Community surveillance enhances *Pseudomonas aeruginosa* virulence during polymicrobial infection. Proceedings of the National Academy of Sciences. 2013;110(3):1059-64.

16. Noto MJ, Burns WJ, Beavers WN, Skaar EP. Mechanisms of pyocyanin toxicity and genetic determinants of resistance in *Staphylococcus aureus*. Journal of bacteriology. 2017;199(17):e00221-17.

17. Biswas L, Biswas R, Schlag M, Bertram R, Götz F. Small-colony variant selection as a survival strategy for *Staphylococcus aureus* in the presence of *Pseudomonas* aeruginosa. Applied and environmental microbiology. 2009;75(21):6910-2.

18. Filkins LM, Graber JA, Olson DG, Dolben EL, Lynd LR, Bhuju S, et al. Coculture of *Staphylococcus aureus* with *Pseudomonas aeruginosa* drives *S. aureus* towards fermentative metabolism and reduced viability in a cystic fibrosis model. Journal of bacteriology. 2015;197(14):2252-64.

19. Fugère A, Lalonde Séguin D, Mitchell G, Déziel E, Dekimpe V, Cantin AM, et al. Interspecific small molecule interactions between clinical isolates of *Pseudomonas* aeruginosa and *Staphylococcus aureus* from adult cystic fibrosis patients. PLoS One. 2014;9(1):e86705.

20. Nguyen AT, Oglesby-Sherrouse AG. Interactions between *Pseudomonas* aeruginosa and *Staphylococcus aureus* during co-cultivations and polymicrobial infections. Applied microbiology and biotechnology. 2016;100:6141-8.

21. Qazi S, Middleton B, Muharram SH, Cockayne A, Hill P, O'Shea P, et al. N-acylhomoserine lactones antagonize virulence gene expression and quorum sensing in *Staphylococcus aureus*. Infection and immunity. 2006;74(2):910-9.

22. DeLeon S, Clinton A, Fowler H, Everett J, Horswill AR, Rumbaugh KP. Synergistic interactions of *Pseudomonas* aeruginosa and *Staphylococcus aureus* in an in vitro wound model. Infection and immunity. 2014;82(11):4718-28. Epub 2014/08/25. doi: 10.1128/IAI.02198-14. PubMed PMID: 25156721.

23. Limoli DH, Whitfield GB, Kitao T, Ivey ML, Davis Jr MR, Grahl N, et al. *Pseudomonas* aeruginosa alginate overproduction promotes coexistence with *Staphylococcus aureus* in a model of cystic fibrosis respiratory infection. MBio. 2017;8(2):e00186-17.

24. Michelsen CF, Khademi SMH, Johansen HK, Ingmer H, Dorrestein PC, Jelsbak L. Evolution of metabolic divergence in *Pseudomonas aeruginosa* during long-term infection facilitates a proto-cooperative interspecies interaction. The ISME journal. 2016;10(6):1323-36.

25. Price CE, Brown DG, Limoli DH, Phelan VV, O’Toole GA. Exogenous alginate protects *Staphylococcus aureus* from killing by *Pseudomonas* aeruginosa. Journal of bacteriology. 2020;202(8):e00559-19.

26. Treffon J, Block D, Moche M, Reiss S, Fuchs S, Engelmann S, et al. Adaptation of *Staphylococcus aureus* to airway environments in patients with cystic fibrosis by upregulation of superoxide dismutase M and iron-scavenging proteins. The Journal of infectious diseases. 2018;217(9):1453-61.

27. Waters CM, Bassler BL. Quorum sensing: cell-to-cell communication in bacteria. Annual review of cell and developmental biology. 2005;21(1):319-46.

28. Barraza JP, Whiteley M. A *Pseudomonas* aeruginosa Antimicrobial Affects the Biogeography but Not Fitness of *Staphylococcus aureus* during Coculture. Mbio. 2021;12(2):e00047-21.

29. Kvich L, Crone S, Christensen MH, Lima R, Alhede M, Alhede M, et al. Investigation of the Mechanism and Chemistry Underlying *Staphylococcus aureus*' Ability to Inhibit *Pseudomonas* aeruginosa Growth in vitro. Journal of Bacteriology. 2022;204(11):e00174-22.

30. Machan Z, Pitt T, White W, Watson D, Taylor G, Cole P, et al. Interaction between *Pseudomonas* aeruginosa and *Staphylococcus aureus*: description of an antistaphylococcal substance. Journal of medical microbiology. 1991;34(4):213-7.

31. Hoffman LR, Déziel E, d'Argenio DA, Lépine F, Emerson J, McNamara S, et al. Selection for *Staphylococcus aureus* small-colony variants due to growth in the presence of *Pseudomonas* aeruginosa. Proceedings of the National Academy of Sciences. 2006;103(52):19890-5.

32. Filipiak W, Sponring A, Baur MM, Filipiak A, Ager C, Wiesenhofer H, et al. Molecular analysis of volatile metabolites released specifically by *Staphylococcus aureus* and *Pseudomonas* aeruginosa. BMC microbiology. 2012;12:1-16.

33. Cendra MdM, Blanco-Cabra N, Pedraz L, Torrents E. Optimal environmental and culture conditions allow the in vitro coexistence of *Pseudomonas* aeruginosa and *Staphylococcus aureus* in stable biofilms. Scientific reports. 2019;9(1):1-17.

34. Gomes-Fernandes M, Gomez A-C, Bravo M, Huedo P, Coves X, Prat-Aymerich C, et al. Strain-specific interspecies interactions between co-isolated pairs of *Staphylococcus aureus* and *Pseudomonas* aeruginosa from patients with tracheobronchitis or bronchial colonization. Scientific Reports. 2022;12(1):3374.

35. Glasser NR, Kern SE, Newman DK. Phenazine redox cycling enhances anaerobic survival in P seudomonas aeruginosa by facilitating generation of ATP and a proton‐motive force. Molecular microbiology. 2014;92(2):399-412.

36. Halsey CR, Lei S, Wax JK, Lehman MK, Nuxoll AS, Steinke L, et al. Amino acid catabolism in *Staphylococcus aureus* and the function of carbon catabolite repression. MBio. 2017;8(1):e01434-16.

37. Rojo F. Carbon catabolite repression in *Pseudomonas*: optimizing metabolic versatility and interactions with the environment. FEMS microbiology reviews. 2010;34(5):658-84.

38. Benjamini Y, Hochberg Y. Controlling the false discovery rate: a practical and powerful approach to multiple testing. Journal of the Royal statistical society: series B (Methodological). 1995;57(1):289-300.